# PROGRESSIVE GAUSSIAN TRANSFORMER WITH ANISOTROPY-AWARE SAMPLING FOR OPEN VOCABULARY OCCUPANCY PREDICTION

**Chi Yan**[1,2] **and Dan Xu**[1*]
[1]The Hong Kong University of Science and Technology (HKUST)
[2]ZEEKR Automobile R&D Co., Ltd
{cyanao,danxu}@cse.ust.hk

## ABSTRACT

The 3D occupancy prediction task has witnessed remarkable progress in recent years, playing a crucial role in vision-based autonomous driving systems. While traditional methods are limited to fixed semantic categories, recent approaches have moved towards predicting text-aligned features to enable open-vocabulary text queries in real-world scenes. However, there exists a trade-off in text-aligned scene modeling: sparse Gaussian representation struggles to capture small objects in the scene, while dense representation incurs significant computational overhead. To address these limitations, we present **PG-Occ**, an innovative **P**rogressive **G**aussian Transformer Framework that enables open-vocabulary 3D occupancy prediction. Our framework employs progressive online densification, a feed-forward strategy that gradually enhances the 3D Gaussian representation to capture fine-grained scene details. By iteratively enhancing the representation, the framework achieves increasingly precise and detailed scene understanding. Another key contribution is the introduction of an anisotropy-aware sampling strategy with spatio-temporal fusion, which adaptively assigns receptive fields to Gaussians at different scales and stages, enabling more effective feature aggregation and richer scene information capture. Through extensive evaluations, we demonstrate that **PG-Occ** achieves state-of-the-art performance with a relative **14.3% mIoU improvement** over the previous best performing method. Code and pretrained models are available at: https://yanchi-3dv.github.io/PG-Occ.

## 1 INTRODUCTION

3D Occupancy perception technology has emerged as a pivotal trend in autonomous driving perception systems, garnering substantial attention from both industry and academia due to its comprehensive perception capabilities (Xu et al., 2025; Zhang et al., 2024). Unlike previous BEV representations (Li et al., 2022), 3D occupancy enriches scene understanding with crucial height information, enabling a complete three-dimensional representation of the environment. Accurate prediction of 3D occupancy and semantic information serves as a cornerstone for robust scene understanding and reconstruction (Cao & de Charette, 2021; Ye & Xu, 2022; Huang et al., 2023). While several benchmarks (Wang et al., 2023; Tian et al., 2024; Sun et al., 2020) have been established to provide semantic annotations for 3D occupancy supervision, they inherently constrain semantic information to predefined categories. This limitation severely hinders the system's ability to perceive general objects beyond these predefined categories.

To enable semantic occupancy detection based on arbitrary user inputs, recent approaches (Tan et al., 2023; Vobecky et al., 2024; Boeder et al., 2024; Zheng et al., 2024a) have shifted away from directly predicting predefined semantic categories of occupancy. Instead, they focus on predicting text-aligned features, which can then be used to calculate similarity scores with text queries to obtain semantic correspondences. This paradigm shift allows open-vocabulary detection capabilities, where the system can identify objects beyond predefined categories by leveraging the rich semantic space of text embeddings. By establishing this text feature alignment in 3D space, these methods effectively bridge the gap between language understanding and spatial perception, enabling more flexible and generalizable occupancy prediction systems. However, due to the high-dimensional na-

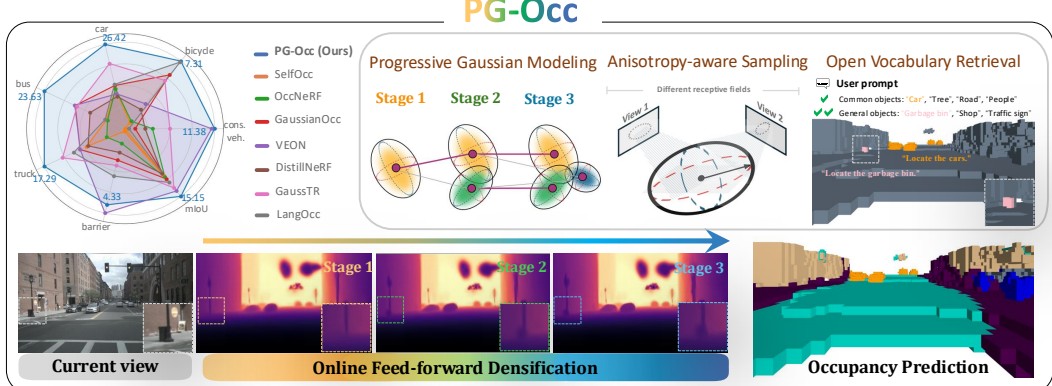

Figure 1: Overview of the proposed PG-Occ framework. The radar chart compares occupancy prediction accuracy across multiple methods, showing the superior performance of PG-Occ. The central panel highlights the key components: progressive Gaussian modeling with online feed-forward densification, anisotropy-aware sampling with adaptive receptive fields, and open-vocabulary retrieval conditioned on prompt inputs. The bottom row illustrates an example progression from the current input view through successive densification stages to the final occupancy prediction.

ture of text features, densely modeling the entire scene incurs substantial memory and computational overhead, severely impacting system efficiency.

Inspired by Gaussian representation (Kerbl et al., 2023) and its applications in perception tasks (Gan et al., 2024; Huang et al., 2024b; Cao et al., 2024), recent work such as GaussTR (Jiang et al., 2024) leverages sparse Gaussian representation to achieve efficient scene perception. However, due to the inherent sparsity of Gaussians, these approaches often struggle to capture fine-grained details in complex scenes, limiting their effectiveness in comprehensive environmental understanding.

To overcome the aforementioned limitations, we introduce **PG-Occ**, a novel Progressive Gaussian Transformer framework for open-vocabulary 3D occupancy prediction. Our framework preserves the computational efficiency of sparse Gaussian representations while overcoming their limitation in modeling fine-grained scene details through an iterative feed-forward densification strategy. Specifically, the framework first leverages coarse base Gaussians to model the global scene structure. It then progressively refines regions with insufficient perception by performing feed-forward densification conditioned on the current prediction. Furthermore, we propose an anisotropy-aware sampling method that selects sample points according to each Gaussian's anisotropy and projects them onto feature planes with varying receptive fields, enabling more effective spatio-temporal feature fusion.

Specifically, we introduce:

- A Progressive Gaussian Transformer framework for open-vocabulary 3D occupancy prediction, which iteratively enhances scene details through online progressive densification guided by perception errors from previous layers, significantly improving perception accuracy.

- An anisotropy-aware sampling method that adaptively adjusts the receptive fields of Gaussians according to their spatial distribution, enabling more effective integration of explicit Gaussian representations with spatio-temporal features.

- Comprehensive experimental validations demonstrating that **PG-Occ** achieves state-of-the-art performance on the challenging Occ3D-nuScenes dataset, with a remarkable relative **14.3% mIoU improvement** over the previous best results.

## 2 RELATED WORK

**Close-set 3D Occupancy Perception.** Using strong close-set 3D labels to supervise 3D occupancy networks is a straightforward idea, and most existing work (Wei et al., 2023; Huang et al., 2023; Zhang et al., 2023b; Ma et al., 2024; Hou et al., 2024) is based on this training approach (Xu et al., 2025). Some improvements focus on efficient spatial representation. SurroundOcc (Wei et al., 2023) extends the BEV with height dimension through spatial cross-attention. TPVFormer (Huang et al., 2023) divides the space into three perspective views, reducing the parameters and computational costs. FastOcc (Hou et al., 2024) accelerates processing by replacing 3D convolutional networks with lightweight 2D BEV convolutions, while GaussianFormer (Huang et al., 2024b) reduces

computation in empty spaces by utilizing sparse Gaussian representations. Another line of research focuses on label efficiency (Pan et al., 2024; Zhang et al., 2023a; Gan et al., 2024; Huang et al., 2024a; Jiang et al., 2024), drawing inspiration from NeRF and 3D Gaussian splatting techniques. These approaches distill 3D occupancy information from Gaussians extracted by 2D foundation models, significantly reducing the need for extensive 3D annotations.

**Open-vocabulary 3D Occupancy Perception.** Current 3D occupancy benchmarks feature categories with varying semantic clarity - some, like "car", "pedestrian", and "truck", have explicit definitions, while others such as "manmade" and "vegetation", remain vague. These broader categories contain numerous undefined semantics that would benefit from finer-grained subdivision to better characterize driving environments. Novel objects are typically classified merely as general obstacles, lacking the flexibility to expand perception based on human prompts (Cao et al., 2023). To address this challenge, OVO (Tan et al., 2023) pioneered a framework that enables open-vocabulary 3D occupancy perception by distilling knowledge from a frozen 2D open-vocabulary segmenter and CLIP text encoder into the 3D model. Similarly, POP-3D (Vobecky et al., 2024) designed a semi-supervised framework incorporating three modalities to improve zero-shot open-vocabulary capabilities. VEON (Zheng et al., 2024a) further advanced the performance by assembling and adapting two complementary 2D foundation models. To reduce dependence on LiDAR sensors, LangOcc (Boeder et al., 2024) integrated Neural Radiance Fields (NeRF), enabling purely vision-based perception approaches. To address the computational overhead of high-dimensional text-vision features, GaussTR (Jiang et al., 2024) models scenes as sparse unstructured Gaussian blobs, achieving reconstruction through a camera-wise feed-forward approach.

**Generalizable 3D Gaussian Splatting with Feed-forward Networks.** A significant limitation of vanilla 3D Gaussian splatting (Kerbl et al., 2023) is its requirement for offline scene-specific optimization rather than efficient feed-forward inference Wang et al. (2025). The success of feed-forward approaches comes from the learning of powerful priors from large datasets. Recent works such as Splatter Image (Szymanowicz et al., 2024) and pixelSplat (Charatan et al., 2024) have proposed novel approaches that enable direct prediction of 3D Gaussian Spaltting parameters from one or two input views, respectively. GPS-Gaussian (Zheng et al., 2024b) utilizes feed-forward networks for human reconstruction. DrivingForward (Tian et al., 2025) introduces this paradigm in sparse-view 3D reconstruction for autonomous driving scenarios. GaussTR (Jiang et al., 2024) conceptualizes the perception of the autonomous driving scene as a task to predict 3D Gaussians from six surround view cameras, while GaussianFlowOcc (Boeder et al., 2025) captures dynamic scenes through Gaussian flow estimation. Unlike existing approaches, our work focuses on progressive Gaussian modeling, employing a coarse-to-fine approach to enhance scene perception capabilities.

## 3 METHODOLOGY

As illustrated in Fig. 2, PG-Occ introduces a novel open-vocabulary 3D occupancy prediction framework. Our approach represents scenes as a set of text-aligned feature Gaussian blobs in a *progressive feed-forward* manner. In Section 3.1, we first present 3D feature Gaussians as an effective scene representation for open-vocabulary occupancy prediction. Subsequently, in Section 3.2, we describe how spatio-temporal image features are progressively transformed into Gaussian representations through an iterative architecture, consisting of a base layer followed by $B$ progressive layers. Additionally, we propose an anisotropy-aware sampling mechanism for spatio-temporal feature fusion, enabling more precise and robust scene understanding. The loss functions used to train our model are detailed in Section 3.3. Finally, as detailed in Section 3.4, we convert the final Gaussian representations into a dense 3D occupancy field.

### 3.1 3D FEATURE GAUSSIAN SPLATTING

Open-vocabulary 3D occupancy prediction aims to identify and localize occupancy regions around a vehicle that correspond to arbitrary text prompts $c_{text}$, given $L$ spatio-temporal camera views $I = \{I_1, ..., I_L\}$ at the current time step. Directly predicting 3D text-aligned voxel features is computationally and memory-intensive due to the high dimensionality of text features. Inspired by 3D Gaussian Splatting (Kerbl et al., 2023; Zhou et al., 2024), we model the driving scene as a set of sparse feature Gaussian blobs $\mathcal{G}$. Vanilla 3D Gaussians typically encode color features; in contrast, we replace them with high-dimensional text-aligned features to better capture semantic information for open-vocabulary occupancy prediction. Formally, each feature Gaussian blob $G_i$

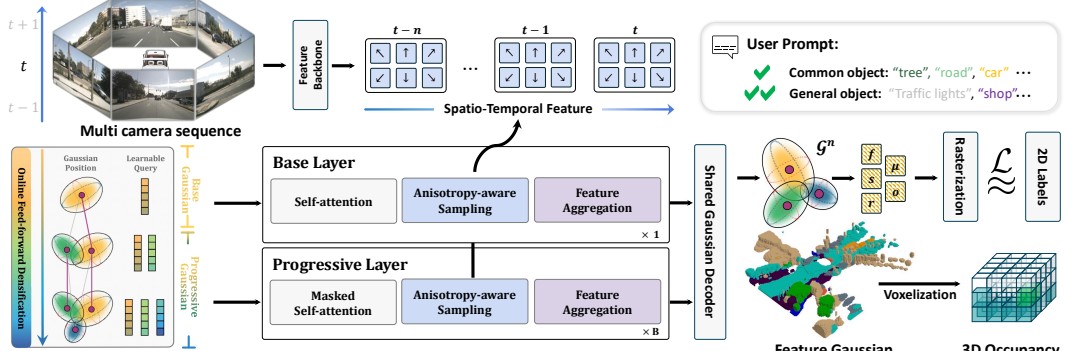

Figure 2: Architecture of the proposed PG-Occ framework. The scene is represented as feature Gaussian blobs, starting from a base layer and progressively refined and densified through $B$ layers. Multi-camera inputs are processed to extract spatio-temporal features, which guide the update and refinement of the Gaussians, which are then voxelized to produce an any-resolution 3D occupancy field, enabling both geometric reconstruction and open-vocabulary semantic understanding.

(hereafter simply called "Gaussian") is defined by its spatial position $\mu_i \in \mathbb{R}^3$, scale $s_i \in \mathbb{R}^3$, rotation quaternion $r_i \in \mathbb{R}^4$, opacity $\sigma_i \in \mathbb{R}$, and a text-aligned feature $f_i \in \mathbb{R}^{512}$:

$$\mathcal{G} = \{G_i : (\mu_i, s_i, r_i, \sigma_i, f_i) \mid i = 1, ..., N\}, \tag{1}$$

where $N$ denotes the number of Gaussian blobs in the scene.

Gaussian representations, in addition to their inherent sparsity, enable efficient rendering and facilitate training via 2D label supervision. Given the camera pose $\mathbf{T}_l$ and intrinsic matrix $\mathbf{K}_l$, the Gaussians $\mathcal{G}$ can be efficiently rasterized onto the 2D camera plane, producing per-pixel expected depth $\hat{D}$ and feature map $\hat{F}$. For each pixel, these values are computed as:

$$\hat{D} = \frac{\sum_{i \in N} d_i \alpha_i \prod_{j=1}^{i-1} (1 - \alpha_j)}{\sum_{i \in N} \alpha_i \prod_{j=1}^{i-1} (1 - \alpha_j)}, \quad \hat{F} = \sum_{i \in N} f_i \alpha_i \prod_{j=1}^{i-1} (1 - \alpha_j), \tag{2}$$

where $d_i$ represents the depth value of the $i$-th 3D Gaussian center point $\mu_i$ projected along the $z$-axis in the camera coordinate system, and $\alpha_i$ denotes the blending weight of the $i$-th Gaussian.

## 3.2 PROGRESSIVE 3D GAUSSIAN MODELING

The core of PG-Occ is a set of learnable and adaptively expandable Gaussian queries $\mathcal{G} \in \mathbb{R}^{N \times D}$ with spatial positions $\mu$, where the latent feature dimension $D$ encodes Gaussian blob attributes, including scale $s$, rotation $r$, opacity $o$, and feature vector $f$. These Gaussian queries are processed by the Progressive Gaussian Transformer, which consists of a base Gaussian layer that captures coarse scene geometry, followed by $B$ progressive layers that iteratively refine the Gaussian representations. An illustrative example of this online progressive refinement is shown in Fig. 1.

Each progressive layer $b$ further incorporates *three core components*: Progressive Online Densification (POD), Asymmetric Self-Attention (ASA), and Anisotropy-aware Feature Sampling (AFS). Prior Gaussian-based occupancy estimation methods (Jiang et al., 2024; Boeder et al., 2025) use a fixed number of $N$ queries, limiting their ability to model complex scenes. In contrast, our approach adaptively expands the queries in each layer by adding $N^b$ queries via a feed-forward module conditioned on the output of the preceding layer $\mathcal{G}^{b-1}$. This progressive expansion allows adaptive, fine-grained modeling of intricate scene structures while maintaining computational efficiency.

### 3.2.1 PROGRESSIVE ONLINE DENSIFICATION (POD)

In contrast to vanilla 3D Gaussian Splatting (Kerbl et al., 2023), which initializes Gaussians from structure-from-motion point clouds and relies on gradient-based densification, we propose an efficient feed-forward strategy for real-time Gaussian densification. Our approach comprises two stages: base initialization, which establishes initial Gaussian positions, and feed-forward densification, which adaptively augments the Gaussian set in regions where the scene remains underrepresented. This design enables online refinement of scene geometry without the computational overhead of gradient backpropagation.

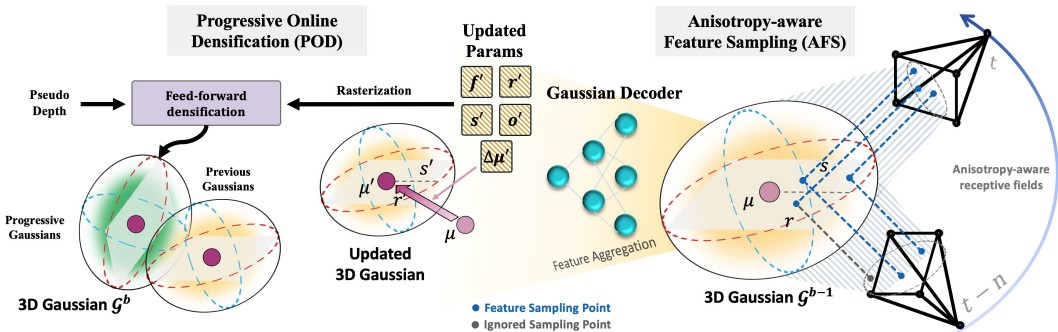

Figure 3: Illustration of the Progressive Online Densification (POD) and Anisotropy-aware Feature Sampling (AFS) modules. POD leverages depth-aware densification to progressively add and refine 3D Gaussians. AFS exploits the anisotropic properties of Gaussians, sampling feature points within anisotropy-aware receptive fields to enable more effective spatio-temporal feature extraction.

**Base Initialization.** To capture coarse scene geometry, we utilize pseudo depth maps from Metric3D V2 (Hu et al., 2024). The downsampled depth map $D$ is back-projected into the ego-vehicle frame using the camera-to-ego transformation $T$ and camera intrinsics $K$, yielding pseudo point clouds $P$:

$$P = \bigcup_{l=1}^{L} T_l \cdot (K_l^{-1} \cdot D_l) \tag{3}$$

Farthest Point Sampling (FPS) (Eldar et al., 1997) is applied to select $N$ representative points as initial Gaussian positions $\mu_i^0$. Each selected point is paired with a trainable query feature, while the Gaussian scale $s_i^0$ is fixed and the rotation $r_i^0$ is initialized as a unit quaternion.

**Feed-forward Densification.** After obtaining the intermediate Gaussian representation $\mathcal{G}^{b-1}$ from the $(b-1)$-th layer (details in Section 3.2.3), we render an expected depth map $\hat{D}$ using Eq. (2). By comparing $\hat{D}$ with the reference depth $D$, we identify under-represented regions:

$$\mathcal{U}_{\text{select}} = \{(u, v) \in \Omega \mid \hat{D}(u, v) - D(u, v) > \gamma\}, \tag{4}$$

where $\gamma$ is set to half the final occupancy grid resolution. This module relies solely on Gaussian rendering, avoiding gradient computation and maintaining efficiency. For each under-represented region, we generate a point set $P^b$ and sample $n^b$ new points via FPS. The new Gaussians $\mu_{\text{add}}^b$ and their query features $q_{\text{add}}^b$ are concatenated with the previous layer's optimized positions $\mu^{b-1}$ and queries $q^{b-1}$ to form the input for the $b$-th transformer layer:

$$\mu^b = \mu^{b-1} \oplus \mu_{\text{add}}^b, \quad q^b = q^{b-1} \oplus q_{\text{add}}^b \tag{5}$$

### 3.2.2 ASYMMETRIC SELF-ATTENTION (ASA)

In progressive Gaussian modeling, newly added Gaussians from online densification are initially under-optimized. Self-attention (Vaswani et al., 2017) is widely employed to model relationships between Gaussians, enhancing overall scene representation. However, applying standard self-attention in this setting risks allowing these under-optimized Gaussians to interfere with the well-trained ones from earlier stages, potentially causing training instability.

To address this, we introduce an Asymmetric Self-Attention (ASA) mechanism that enforces asymmetric interactions: newly added Gaussians cannot influence the existing, well-optimized Gaussians, while they can attend to and leverage the features of the existing ones to refine their own under-optimized representations. This design ensures that previously learned Gaussians remain stable, while the newly extended Gaussians progressively improve by utilizing existing information.

Formally, let $x_b$ denote the number of Gaussian queries in layer $b$, with the first $x_{b-1}$ inherited from the previous layer and the remaining $x_b - x_{b-1}$ newly added. Given Gaussian queries $q^b$ and their positional encodings $\text{PE}(\mu^b)$, the ASA operation is defined as:

$$q_{asa}^b = \text{ASA}(q^b + \text{PE}(\mu^b), \ q^b + \text{PE}(\mu^b), \ q^b, \ M), \tag{6}$$

where PE denotes the positional encoding. The attention mask $M \in \mathbb{R}^{x_b \times x_b}$ is constructed as:

$$M_{i,j} = \begin{cases} -\infty, & \text{if } i < x_{b-1} \text{ and } j \geq x_{b-1} \\ 0, & \text{otherwise} \end{cases} \tag{7}$$

By restricting the influence of new Gaussians, ASA stabilizes progressive Gaussian modeling while enabling effective inter-Gaussian feature propagation.

### 3.2.3 ANISOTROPY-AWARE FEATURE SAMPLING (AFS)

Treating each Gaussian $G_i$ solely as a point at its center $\mu_i$ oversimplifies feature sampling and ignores the anisotropic properties encoded by its scale $s$ and rotation quaternion $r$, which significantly affect the Gaussian's effective receptive field in 2D feature space. This simplification reduces sampling accuracy, as the anisotropic geometry strongly influences the effective receptive field of each Gaussian in the 2D feature space, as illustrated in Fig. 3. To exploit anisotropy, the query feature $q_{asa}$ is passed through an MLP to generate $n$ unit offsets $\mu_\delta$ (with $n = 16$ in practice), which are scaled and rotated to lie within the Gaussian ellipsoid:

$$\mu_{\text{sample}}^{i,j} = \mu_i + R(r_i) \cdot (s_i \odot \mu_\delta^{i,j}), \quad j = 1, \ldots, n \tag{8}$$

where $R(r)$ is the rotation matrix derived from the quaternion $r$, $s$ is the Gaussian scale vector, and $\odot$ denotes element-wise multiplication.

The resulting 3D sampling points $\mu_{\text{sample}}^{i,j}$ are projected onto 2D feature planes using known camera intrinsics and extrinsics. To aggregate the corresponding features across views and timestamps, we adopt a simple yet effective feature aggregation module from prior work (Liu et al., 2023):

$$f_a^i = \text{Aggregation}\Big(\{\text{Interp}(\mu_{\text{sample}}^{i,j}, F)\}_{j=1}^n\Big), \tag{9}$$

where $\text{Interp}(\cdot)$ denotes bilinear interpolation at the projected 2D locations, and $\{\cdot\}_{j=1}^n$ indicates the collection of $n$ sampled features corresponding to the $i$-th Gaussian, which are subsequently aggregated to form a single representative feature.

Finally, the aggregated feature $f_a$ is fed into two lightweight MLPs to decode geometric properties and text-aligned features separately:

$$f_i = \text{MLP}_{feat}(f_a^i), \quad (\Delta\mu_i, s_i, r_i, \sigma_i) = \text{MLP}_{geo}(f_a^i), \quad \mu_i = \mu_{i-1} + \Delta\mu_i, \tag{10}$$

where $\Delta\mu_i$ represents a learned displacement, allowing the model to refine Gaussian positions incrementally rather than predicting absolute coordinates directly.

### 3.3 EFFICIENT TRAINING VIA RASTERIZATION

Due to the lack of labeled open-vocabulary 3D occupancy data, our model is trained using only 2D supervision. Specifically, we leverage text-aligned features $F$ and pseudo depth maps $D$, both of which are extracted directly from images. This approach eliminates the need for LiDAR scans or any explicit 3D point cloud data. To ensure stable training, we rasterize the 3D Gaussians $\mathcal{G}_b$ from each layer onto the 2D image plane and supervise the model using the losses described below.

**Depth Rendering Loss.** The depth rendering loss combines SILog, L1, and temporal photometric consistency losses (Godard et al., 2019; Yao et al., 2024) for geometric consistency:

$$\mathcal{L}_{depth} = \mathcal{L}_{L1}(D, \hat{D}) + \lambda_{SILog}\mathcal{L}_{SILog}(D, \hat{D}) + \lambda_{temp}\mathcal{L}_{temp}(D, \hat{D}), \tag{11}$$

where $\lambda_{SILog}$, and $\lambda_{temp}$ are weighting coefficients to balance different depth loss terms.

**Feature Rendering Loss.** The feature rendering loss combines mean squared error (MSE) loss and cosine similarity loss to achieve the feature alignment:

$$\mathcal{L}_{feat} = \mathcal{L}_{cos}(F, \hat{F}) + \lambda_{mse}\mathcal{L}_{mse}(F, \hat{F}), \tag{12}$$

where $\lambda_{mse}$ represents a weighting coefficient to balance the cosine similarity and MSE losses.

**Final Objective.** The final loss combines the depth and feature rendering losses as follows:

$$\mathcal{L}_{total} = \lambda_{depth}\mathcal{L}_{depth} + \lambda_{feat}\mathcal{L}_{feat}, \tag{13}$$

where $\lambda_{depth}$ and $\lambda_{feat}$ are weighting coefficients to balance the two losses.

### 3.4 TEST-TIME INFERENCE VIA VOXELIZATION

After obtaining the progressive 3D Gaussian representation of the scene based on the current camera input, we use the decoded text-aligned feature Gaussian blobs of the final transformer layer for

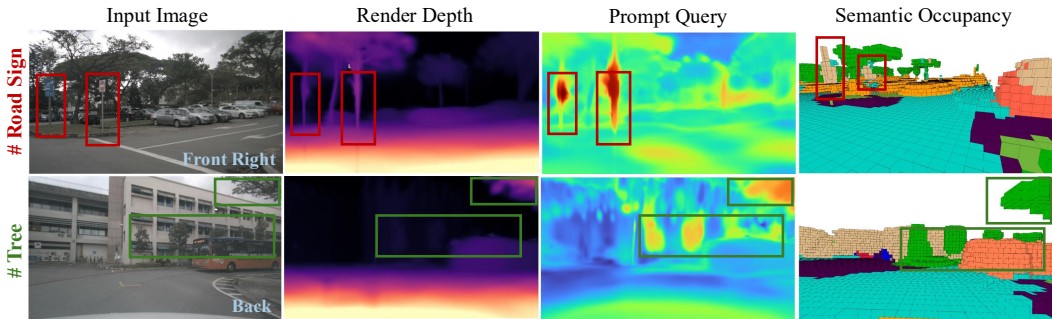

Figure 4: Illustration of PG-Occ predictions. Given camera inputs and text prompts, the method predicts depth (column 2), produces open-vocabulary semantic labels (column 3), and generates the final semantic occupancy map (column 4). Additional visualizations are provided in appendix C.1.

evaluation. We convert feature Gaussians into semantic occupancy via a two-step process. First, arbitrary text prompts $c_{text}$ are encoded using the CLIP text encoder to obtain feature embeddings $f_{text}$, which are then matched against the Gaussian features to assign semantic labels to 3D Gaussians. Second, a Gaussian-to-voxel post-processing step transforms the labeled 3D Gaussians into a dense occupancy representation. In particular, since our method does not require dense occupancy labels during training, this conversion is applied only at inference. Further technical details on the text prompt and the Gaussian-to-voxel module are provided in appendix D.1.

## 4 EXPERIMENTS

### 4.1 EXPERIMENTAL SETUP

**Datasets and Metrics.** We conduct comprehensive experiments on two benchmarks: Occ3D-nuScenes and nuScenes retrieval. *Open-vocabulary occupancy prediction* is evaluated on the Occ3D-nuScenes dataset (Tian et al., 2024), which contains 1,000 scenes captured by surround-view cameras and LiDAR sensors. Semantic labels are assigned to each voxel based on predefined text queries (see appendix D.2). Performance is measured using IoU, mean IoU (mIoU), and ray-IoU metrics. *Open-vocabulary occupancy retrieval* is conducted on the nuScenes retrieval benchmark proposed by POP-3D (Vobecky et al., 2024), which contains 105 samples. Each LiDAR point cloud is paired with a text query, and retrieval performance is evaluated using mean average precision (mAP) over all LiDAR points, as well as mAP (v), which considers only points visible in at least one camera. *Depth estimation* follows the GaussianOcc (Gan et al., 2024) setting, where the ground truth depth maps are obtained by projecting LiDAR point clouds. Standard error metrics are used for evaluation, including absolute relative error (Abs Rel), square relative error (Sq Rel), root mean square error (RMSE), and RMSE log.

**Implementation Details.** We adopt ResNet-50 (He et al., 2016) as our image feature extraction backbone, utilizing the previous seven frames to capture spatio-temporal information. Our Progressive Gaussian Transformer comprises one base layer and two progressive layers. All experiments are run on 8 × A800 GPUs, with 8 epochs of training (approximately 9 hours). To improve computational efficiency, we use a resolution of 180 × 320 for depth and feature rasterization, as well as Gaussian point initialization. More implementation details can be found in appendix D.

### 4.2 MAIN EXPERIMENT RESULTS

**Semantic Occupancy Prediction Results.** We report open-vocabulary occupancy prediction results on the Occ3D-nuScenes dataset (Tian et al., 2024) in Table 1 and group methods based on the sensor modalities employed during training (Camera, LiDAR, and Text). To ensure a fair comparison and emphasize the benefits of our approach, LangOcc, GaussTR, and our PG-Occ all use MaskCLIP (Zhou et al., 2022) for text supervision. Our method achieves SOTA performance with an mIoU of **15.15**, corresponding to a 14.3% relative improvement over previous best methods. Remarkably, despite not using LiDAR data during training, PG-Occ outperforms VEON and other competitors. As shown in the LiDAR chart in Fig. 1, our approach excels at detecting medium-sized objects. The slightly lower performance on small objects can be attributed to the coarse voxel resolution, with a voxel size of 0.4 m, which limits the contribution of finely optimized Gaussians.

Table 1: Quantitative performance of 3D occupancy methods on **Occ3D-nuScenes** dataset. The *Mod.* column specifies the sensor/modalities used for training: C for Camera, L for LiDAR, and T for Text. IoU scores for "others" and "other flat" classes are consistently zero and thus omitted. "Cons veh." means construction vehicles, and "drive. surf." means drivable surfaces. The best and second-best results are denoted in **bolded** and underlined, respectively.

| Method | Mod. | mIoU | barrier | bicycle | bus | car | cons. veh. | motorcycle | pedestrian | traffic cone | trailer | truck | drive. surf. | sidewalk | terrain | manmade | vegetation |
|---|---|---|---|---|---|---|---|---|---|---|---|---|---|---|---|---|---|
| SelfOcc (Huang et al., 2024a) | C | 10.54 | 0.15 | 0.66 | 5.46 | 12.54 | 0.00 | 0.80 | 2.10 | 0.00 | 0.00 | 8.25 | 55.49 | 26.30 | 26.54 | 14.22 | 5.60 |
| OccNeRF (Zhang et al., 2023a) | C | 10.81 | 0.83 | 0.82 | 5.13 | 12.49 | 3.50 | 0.23 | 3.10 | 1.84 | 0.52 | 3.90 | 52.62 | 20.81 | 24.75 | 18.45 | 13.19 |
| GaussianOcc (Gan et al., 2024) | C | 11.26 | 1.79 | 5.82 | 14.58 | 13.55 | 1.30 | 2.82 | 7.95 | 9.76 | 0.56 | 9.61 | 44.59 | 20.10 | 17.58 | 8.61 | 10.29 |
| GaussianFlowOcc (Boeder et al., 2025) | C | 14.07 | 6.27 | 8.54 | 13.36 | 12.38 | 4.92 | 10.05 | 6.84 | 8.75 | 1.12 | 10.43 | 54.40 | 26.44 | 28.89 | 10.39 | 9.33 |
| VEON (Zheng et al., 2024a) | C+L+T | 13.95 | 4.80 | 2.70 | 14.70 | 10.90 | 11.00 | 3.80 | 4.70 | 4.00 | 5.30 | 9.60 | 46.50 | 21.10 | 22.10 | 24.80 | 23.70 |
| DistillNeRF (Wang et al., 2024) | C+L+T | 10.05 | 1.35 | 2.08 | 10.21 | 10.09 | 2.56 | 1.98 | 5.54 | 4.62 | 1.43 | 7.90 | 43.02 | 16.86 | 15.02 | 14.06 | 15.06 |
| LangOcc (Boeder et al., 2024) | C+T | 12.04 | 2.70 | 7.20 | 5.80 | 13.90 | 0.50 | **10.8** | **6.40** | **8.70** | **3.20** | 11.00 | **42.10** | 12.50 | **27.20** | 14.10 | 14.50 |
| GaussTR (Jiang et al., 2024) | C+T | 13.25 | 2.09 | 5.22 | 14.07 | 20.43 | 5.70 | 7.08 | 5.12 | 3.93 | 0.92 | 13.36 | 39.44 | 15.68 | 22.89 | **21.17** | 21.87 |
| PG-Occ (Ours) | C+T | **15.15** | **4.33** | **7.31** | **23.63** | **26.42** | **11.38** | 6.33 | 2.74 | 5.79 | 3.07 | **17.29** | 37.81 | **19.29** | 20.85 | 19.02 | **21.92** |

Table 2: Depth estimation error metrics on the nuScenes validation set. The best results denoted in **bold**. Abs Rel is used as the primary evaluation metric.

| Method | Abs Rel ↓ | Sq Rel ↓ | RMSE ↓ | RMSE log ↓ |
|---|---|---|---|---|
| SelfOcc (Huang et al., 2024a) | 0.215 | 2.743 | 6.706 | 0.316 |
| OccNeRF (Zhang et al., 2023a) | 0.202 | 2.883 | 6.697 | 0.319 |
| GaussianOcc (Gan et al., 2024) | 0.197 | 1.846 | 6.733 | 0.312 |
| GaussianFlowOcc (Boeder et al., 2025) | 0.278 | 2.522 | 5.232 | 0.389 |
| Metric3D V2 (Hu et al., 2024) | 0.170 | 4.016 | 6.453 | 0.291 |
| PG-Occ (Ours) | **0.139** | **1.159** | **5.466** | **0.269** |

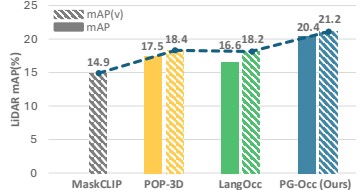

Figure 5: SOTA comparison on the **nuScenes retrieval** dataset.

**nuScenes Retrieval Dataset Results.** As shown in Fig. 5, our method achieves a visible mean Average Precision (mAP(v)) of **21.2** on the nuScenes retrieval dataset. This outperforms the existing vision-based method, LangOcc, which achieves 18.2. This improvement demonstrates the effectiveness of progressive Gaussian modeling in enhancing both accuracy and robustness for 3D open-vocabulary retrieval tasks. Note that GaussTR does not report results on the nuScenes retrieval dataset and cannot be evaluated using LiDAR points. Qualitative results in Fig. 1 further illustrate our system's language-based retrieval capabilities. Queries such as "Locate the cars" and "Locate the garbage bin" are precisely grounded within the predicted 3D occupancy grid.

**Depth Estimation Results.** We evaluate the geometric accuracy of our Gaussian scene representation via quantitative depth estimation, presented in Table 2, accompanied by depth visualizations in Fig. 9. Remarkably, although supervision is derived from depth maps of Metric3D V2 (Hu et al., 2024), our model produces depth estimates that exceed the original labels, achieving an error of 0.139, corresponding to an +18.2% boost. This improvement results from geometric constraints imposed by multi-view depth consistency and feature coherence, which help maintain smooth and accurate scene geometry even in challenging regions.

**Additional Corner-case Examples.** As shown in Fig. 8, we present additional examples of open-vocabulary retrieval results for uncommon categories, such as mailboxes and warning signs. Although these categories rarely appear in the dataset, our method is still able to detect them reliably. These examples further demonstrate PG-Occ's ability to generalize to open-vocabulary scenarios, accurately capturing semantic occupancy even for rarely seen or atypical objects.

**Efficiency Comparison.** We compare mIoU, training time, and inference speed (frames per second) to provide a comprehensive assessment of the efficiency of our method. As shown in Table 4, our approach achieves significant relative improvements on the Occ3D-nuScenes dataset (Tian et al., 2024), with a +14.3% increase in mIoU, a +41.1% boost in FPS, and a 25% reduction in training time.

Table 3: Effects of the proposed key model components (i.e., POD, AFS, ASA).

| Method | mIoU | RayIoU | mAP (v) |
|---|---|---|---|
| w/o POD | 14.84 | 12.58 | 19.21 |
| w/o AFS | 15.03 | 13.56 | 20.12 |
| w/o SA | 11.14 | 10.44 | 15.60 |
| w/o ASA | 14.85 | 12.76 | 19.41 |
| Full model | **15.15** | **13.92** | **21.20** |

**Zero-shot Generalization to Unseen Domains.** As shown in Fig. 13, to further validate the generalization capability of our method, we include additional qualitative results on the Lyft Level-5 dataset (Christy et al., 2019). Importantly, we do not retrain or fine-tune our model on the Lyft Level-

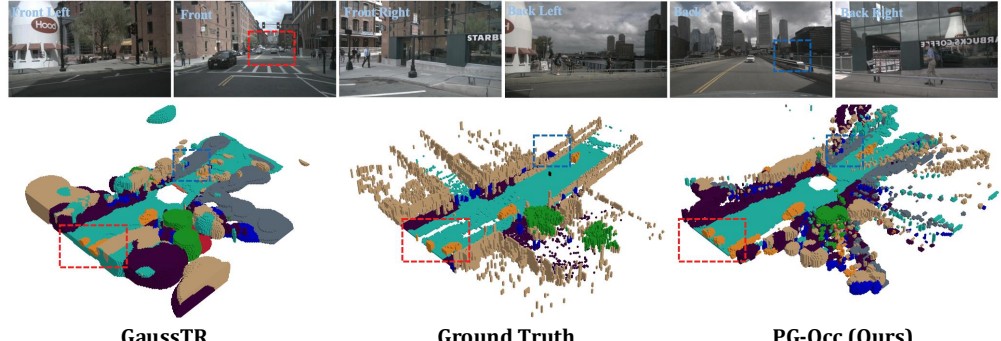

**GaussTR**                    **Ground Truth**                    **PG-Occ (Ours)**

Figure 6: Qualitative comparison of 3D occupancy prediction using PG-Occ and prior methods. This figure presents a visual comparison between PG-Occ (ours), GaussTR, and Ground Truth data in reconstructing urban scenes. PG-Occ achieves more accurate and perceptually coherent 3D occupancy predictions, capturing finer structural details and producing thicker, more realistic surfaces. The red and blue bounding boxes highlight regions where PG-Occ notably outperforms previous SOTA fixed-query methods, demonstrating improved fidelity and spatial consistency.

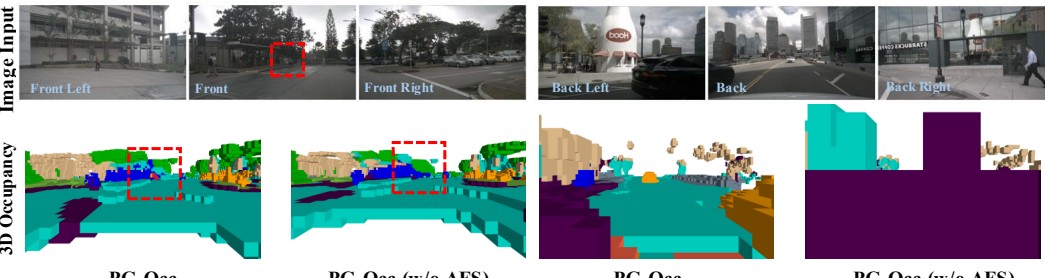

**PG-Occ**       **PG-Occ (w/o AFS)**       **PG-Occ**       **PG-Occ (w/o AFS)**

Figure 7: Qualitative comparison illustrating the effect of Anisotropy-aware Feature Sampling (AFS). The red boxes in the left-side example show that AFS captures finer, semantically-aware details, leading to clearer and more accurate occupancy predictions. The right-side examples further demonstrate that treating anisotropic Gaussian features as identical points (w/o AFS) may result in oversized Gaussians near the camera, which destabilize the overall occupancy estimation. In contrast, our AFS-enhanced model maintains stable and detailed geometric structures.

5 dataset. Instead, we directly perform inference using the model trained solely on nuScenes. This setting represents a challenging domain shift: image resolution, camera positions, camera intrinsics, and the overall environmental domain all differ significantly from those in nuScenes. Despite these differences, our method consistently produces reliable open-vocabulary occupancy predictions, and notably, it is still able to detect and recover small or rare objects in these unseen scenes.

**Robustness Evaluation with Different Pretrained Depth Models.** To further evaluate the robustness of our approach under different depth estimators, we train and test PG-Occ using UniDepth V2 (Piccinelli et al., 2025). Table 13 reports both the depth estimation errors and the open-vocabulary semantic occupancy performance on the nuScenes validation set. The results of additional analyses can be found in appendix B.4.

## 4.3 MAIN ABLATION STUDY

**Effect of Progressive Online Densification (POD).** As visualized in Fig. 1, the Progressive Online Densification module iteratively expands Gaussian queries during inference, allowing the model to refine complex scene geometries progressively. This adaptive densification mechanism dynamically allocates computational resources to regions with insufficient reconstruction, thereby improving the fidelity of recovered details. Compared to prior methods like GaussTR (Jiang et al., 2024), which rely on a fixed number of queries, our approach more effectively captures thick scene surfaces and reconstructs objects with greater precision, as further demonstrated in Fig. 7. The impact of POD is quantitatively validated in the ablation study reported in Table 3, where removing POD leads to substantial performance degradation across all key metrics. These results highlight the crucial role of POD in achieving high-accuracy 3D perception by adaptively focusing model capacity on challenging regions of the scene.

**Effect of Anisotropy-aware Feature Sampling (AFS).** To evaluate the impact of the AFS design, we disable Gaussian anisotropy and treat Gaussians as simple point clouds during sampling. As

Table 4: Efficiency comparison of methods on mIoU, training time, and inference FPS.

| Method | mIoU | Training | FPS |
|---|---|---|---|
| LangOcc (Boeder et al., 2024) | 12.04 | >48 hours | 1.70 |
| GaussTR (Jiang et al., 2024) | 13.25 | 12 hours | 1.04 |
| PG-Occ (6000 Queries) | **15.15** | **9 hours** | **2.40** |

Table 5: Ablation study on the number of extended queries in the progressive layer.

| Queries | mIoU | RayIoU | mAP (v) |
|---|---|---|---|
| 0 | 14.84 | 12.58 | 19.21 |
| 500 | 14.95 | 13.27 | 20.42 |
| 1000 | **15.15** | **13.92** | 21.20 |
| 2000 | 14.79 | 13.21 | **21.29** |

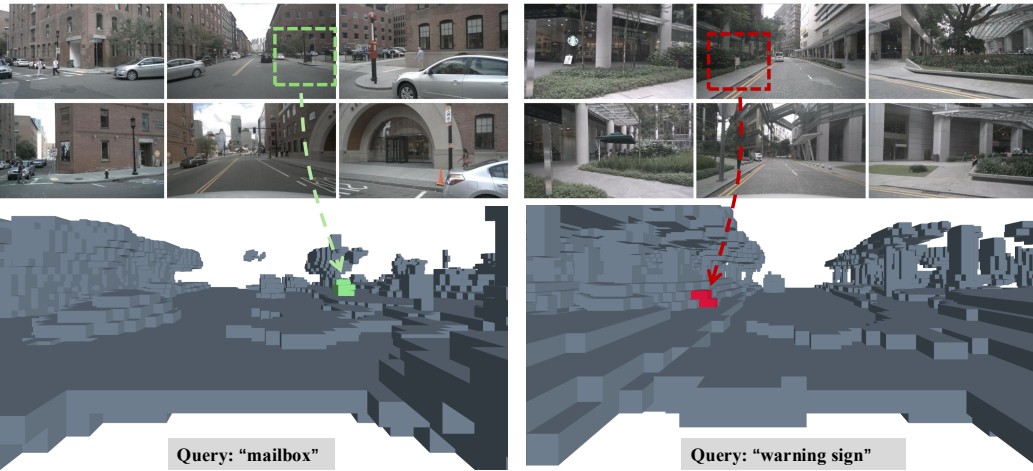

Figure 8: Additional corner-case examples illustrating the capability of PG-Occ for open-vocabulary occupancy retrieval.

shown in Table 3, this results in a 0.12% drop in mIoU. The performance difference stems from the anisotropy strategy, which enables more effective multi-view feature capture, improving both semantic occupancy prediction (mIoU, RayIoU) and open-vocabulary retrieval (mAP(v)).

**Effect of Asymmetric Self-attention Module (ASA).** As shown in Table 3, removing all self-attention ("w/o SA") leads to a substantial performance collapse (mIoU drops from 15.15 to 11.14), indicating that attention-based interactions are essential for maintaining coherent Gaussian features. Reintroducing standard symmetric self-attention ("w/o ASA") significantly alleviates this issue, yet our asymmetric design still delivers the best results, further improving mIoU from 14.85 to 15.15. This improvement suggests that ASA not only enables cross-Gaussian communication but also more effectively preserves historical features while reducing interference from newly added Gaussians during densification, ultimately leading to more stable and accurate feature aggregation.

**Effect of the Number of Extended Gaussian Queries.** We evaluate the impact of varying the number of extended Gaussian queries, keeping the base queries fixed at 4000. As shown in Table 5, increasing extended queries gradually improves the mIoU from 14.84 to 15.15, although a slight drop occurs at 2000 queries. We attribute this to the fact that both the mIoU and RayIoU metrics are evaluated at a voxel size of 0.4, where further Gaussian refinement provides limited gains for scene representation. However, the mAP(v) metric evaluated on LiDAR data, not constrained by voxel resolution, continues to improve at 2000 queries, achieving an increase of 21.29, indicating that a larger number of extended Gaussian queries still benefits scene optimization.

**Additional Ablation Studies.** We provide further analyses in appendix B.1, covering model design (base Gaussian queries, densification threshold, and number of sampling points), training supervision (loss terms and photometric cues), and efficiency/robustness (layer-wise time and pose noise).

## 5 CONCLUSION AND LIMITATIONS

In this paper, we propose a progressive Gaussian transformer framework for the open-vocabulary occupancy prediction task. Our method models driving scenes as extendable feature Gaussian blobs in a purely feed-forward manner, achieving state-of-the-art results with high efficiency. The anisotropy-aware sampling also further improves detail capture. However, due to sparse viewpoints in driving scenarios, constraining the Gaussian scale in depth is challenging, which can cause popping artifacts. Additionally, as Gaussians increase during modeling, memory and computation costs grow, potentially affecting real-time performance. For future work, we will explore 4D Gaussian approaches and multi-view constraints to address these issues.

**Acknowledgements.** The research is supported in part by Early Career Scheme of the Research Grants Council (RGC) of the Hong Kong SAR under grant No. 26202321, Department of Science & Technology of Shandong Province under grant No. SDST26EG01, SAIL Research Project, HKUST-Zeekr Coolaborative Research Fund, HKUST-WeBank Joint Lab Project, and Tencent Rhino-Bird Focused Research Program.

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

# PROGRESSIVE GAUSSIAN TRANSFORMER WITH ANISOTROPY-AWARE SAMPLING FOR OPEN VOCABULARY OCCUPANCY PREDICTION

## SUPPLEMENTARY MATERIAL

This supplementary material offers more detailed descriptions to ensure reproducibility, along with extensive evaluations and diverse qualitative results, which collectively highlight the effectiveness, robustness, and efficiency of our proposed method, **PG-Occ**.

▷ **appendix A**: Video demonstrations comparing the open-vocabulary occupancy inference results of PG-Occ, the previous state-of-the-art method, and the Ground Truth on the Occ3D-nuScenes validation and test sets.

▷ **appendix B**: Additional experimental results, including extended quantitative comparisons and additional ablation studies.

▷ **appendix C**: More qualitative visualization results of our PG-Occ.

▷ **appendix D**: Additional details on implementation.

## A  VIDEO DEMONSTRATION

We provide video demonstrations of our open-vocabulary occupancy inference results on the Occ3D-nuScenes (Tian et al., 2024) validation and test sets. For better visualization quality, please refer to our project page (https://yanchi-3dv.github.io/PG-Occ/).

## B  ADDITIONAL EXPERIMENT RESULTS

### B.1  ADDITIONAL ABLATION STUDY

**Impact of the Number of Base Gaussian Queries.** We evaluate the effect of varying the number of base Gaussian queries while keeping the number of extended Gaussian queries fixed at 1000. As shown in Table 6, increasing the number of base layer queries from 1000 to 4000 consistently improves the mIoU from 13.17 to 15.15, indicating enhanced perception accuracy. However, further increasing the queries to 8000 results in a slight drop in mIoU. This decline is due to the excessive number of Gaussian queries overwhelming the self-attention mechanism, thereby weakening the model's ability to capture the critical spatial interactions between queries, similar to the observations reported in Jiang et al. (2024).

Table 6: Ablation study on the number of initial queries in the base layer. The best and the second-best performances of each metric are highlighted with **bold** and underlined in the table.

| Queries | mIoU | RayIoU | mAP (v) |
|---|---|---|---|
| 1000 | 13.17 | 10.33 | 17.25 |
| 2000 | 14.54 | 12.84 | 18.98 |
| 4000 | **15.15** | **13.92** | **21.20** |
| 8000 | 14.99 | 13.52 | 21.07 |

**Ablation Study of Feed-forward Densification Module Threshold.** We investigate the impact of the threshold in the feed-forward densification module, which determines the minimum distance from points outside to the center of an occupancy cell, on both computational cost and prediction accuracy. Gaussian points are selected according to different thresholds, and the subsequent farthest point sampling (FPS) time, mIoU are measured. The results

Table 7: Impact of the feed-forward densification on farthest point sampling (FPS) time and occupancy prediction.

| Threshold | Total FPS Time (ms) | mIoU |
|---|---|---|
| 0.0 | 50 | 15.11 |
| 0.2 | 34 | 15.15 |
| 0.4 | 30 | 15.13 |

are summarized in Table 7. As shown in the table, decreasing the threshold selects more points, increasing the subsequent FPS time from 30 ms to 50 ms, which results in higher computational overhead. Meanwhile, a threshold of 0.2 achieves the highest mIoU of 15.15, slightly outperforming thresholds 0.0 and 0.4 (15.11 and 15.13, respectively). This indicates that the chosen threshold

Table 9: Ablation on the loss function $\mathcal{L}$. The best performances of each metric are highlighted with **bold** in the table.

| $\mathcal{L}_{L1}$ | $\mathcal{L}_{SILog}$ | $\mathcal{L}_{temp}$ | $\mathcal{L}_{mse}$ | $\mathcal{L}_{cos}$ | mIoU | RayIoU | mAP (v) |
|---|---|---|---|---|---|---|---|
| ✓ | ✓ | ✓ | ✓ |   | 13.51 | 12.30 | 18.13 |
| ✓ | ✓ | ✓ |   | ✓ | 15.12 | 13.81 | 20.52 |
| ✓ | ✓ |   | ✓ | ✓ | 14.47 | 13.01 | 19.14 |
| ✓ |   | ✓ | ✓ | ✓ | 15.10 | 13.89 | 20.41 |
|   | ✓ | ✓ | ✓ | ✓ | 14.69 | 13.23 | 19.95 |
| ✓ | ✓ | ✓ | ✓ | ✓ | **15.15** | **13.92** | **21.20** |

of 0.2 effectively balances point selection for Gaussian densification, maintaining high prediction accuracy while controlling computational cost.

**Ablation Study of the Number of Sampling Offsets.** We evaluate the impact of the number of sampling offsets per Gaussian on occupancy prediction. As shown in Table 8, increasing the number of offsets from 8 to 32 consistently improves performance, with mIoU rising from 15.05 to 15.46. This demonstrates that a larger number of sampling points allows the network to capture more fine-grained scene details, enhancing occupancy prediction accuracy. However, the increase in offsets also leads to longer training times, growing from 8 hours for 8 offsets to 11.2 hours for 32 offsets on an 8×A800 GPU setup. These results highlight a trade-off between accuracy and computational cost, indicating that 16 offsets provide a balanced choice, achieving strong performance with moderate training time.

**Ablation Study of Loss.** We systematically evaluate the impact of different loss function combinations, including $\mathcal{L}_{L1}$, $\mathcal{L}_{SILog}$, $\mathcal{L}_{temp}$, $\mathcal{L}_{mse}$, and $\mathcal{L}_{cos}$. The results are summarized in Table 9. We observe that using all five loss functions consistently yields the best performance, achieving a mIoU of 15.15%, a RayIoU of 13.92%, and an mAP of 21.20%. Omitting any individual loss results in a slight drop across these metrics, indicating that each component contributes to both geometric accuracy and feature alignment. These findings confirm that the combination of complementary losses enables PG-Occ to more effectively capture fine-grained scene details and improve overall 3D occupancy prediction.

Table 8: Impact of the number of sampling offsets per Gaussian on performance and training time.

| Sampling Points | mIoU | Training Time |
|---|---|---|
| 8 | 15.05 | 8 hours |
| 16 | 15.15 | 9 hours |
| 32 | 15.46 | 11.2 hours |

**Ablation Study of Photometric Supervision.** Our occupancy prediction aims to recover both geometric and semantic components. Due to the challenges of large-scale outdoor scenes and limited view supervision, photometric information often fails to provide effective geometric supervision. Moreover, color features do not reliably correspond to semantic categories in such scenes, so we exclude them during training. We performed an ablation study by adding a photometric prediction head to regress 3D color values for supervision. The quantitative results are summarized in Table 10.

Table 10: Ablation study on photometric supervision. The best performances of each metric are highlighted with **bold**.

| Color Supervision | mIoU | RayIoU |
|---|---|---|
| w/o color supervision | **15.15** | **13.92** |
| w/ color supervision | 14.96 | 13.89 |

**Ablation Study of Pose Noise.** We investigate the robustness of PG-Occ to pose noise during temporal-spatial feature fusion by adding Gaussian perturbations with different standard deviations to the historical ego poses during inference, as summarized in Table 11. The results show that a slight amount of pose noise leads to a minor improvement in mIoU, while larger noise levels cause a small decrease followed by stabilization, demonstrating that PG-Occ is robust to pose errors. We attribute this robustness to two factors: first, the nuScenes dataset poses are not perfectly accurate and contain small misalignments, which naturally provide tolerance to minor noise; second, significant pose inaccuracies result in feature sampling failures on the camera plane, preventing unreliable features from degrading system performance and thereby maintaining relatively high perception accuracy.

Table 11: Impact of Gaussian pose noise on occupancy prediction.

| Standard Deviation | 0 | 0.01 | 0.1 | 0.5 |
|---|---|---|---|---|
| mIoU | 15.15 | 15.19 | 15.12 | 15.12 |

## B.2 LAYER-WISE TIME CONSUMPTION

The Table 12 reports the inference time of each Gaussian transformer layer in milliseconds. The base layer, using the fewest Gaussians, achieves the fastest speed at 27.4 ms. As more Gaussians are added in the First and Second Progressive Layers, the inference time correspondingly increases to 58.3 ms and 60.6 ms, reflecting the higher computational cost of processing denser representations.

Table 12: Inference time of different layers.

| Component | Time (ms) |
|---|---|
| Base Layer | 27.4 |
| First Progressive Layer | 58.3 |
| Second Progressive Layer | 60.6 |

## B.3 ROBUSTNESS EVALUATION WITH DIFFERENT PRETRAINED DEPTH MODELS

In the main paper, we adopt Metric3D V2 (Hu et al., 2024) as our default depth estimator. To further examine the robustness of PG-Occ under different sources of depth supervision, we additionally train and evaluate our model using UniDepth V2 (Piccinelli et al., 2025) pseudo-depth. The Table 13 reports both the depth estimation errors and open-vocabulary semantic occupancy performance on the nuScenes validation set.

Similar to the setting with Metric3D V2 pseudo-depth, using UniDepth V2 also enables PG-Occ to recover depth estimates that outperform the pseudo-labels. For example, while UniDepth V2 provides an Abs Rel of 0.158, PG-Occ improves it to 0.137, showing that PG-Occ can consistently refine imperfect depth supervision regardless of the depth model used.

In terms of the open-vocabulary occupancy prediction task, the performance remains stable across depth models. When switching from Metric3D V2 to UniDepth V2 pseudo-depth, the mIoU only changes slightly from 15.15 to 15.08, confirming that PG-Occ is largely insensitive to the specific choice of depth estimator.

These findings highlight two key properties of PG-Occ. **(i)** PG-Occ does not depend on any specific depth architecture; it only requires coarse geometric cues for initialization and supervision, making it naturally compatible with diverse metric depth models. **(ii)** PG-Occ consistently refines these cues and maintains robust occupancy performance even as the upstream depth model changes.

Table 13: Robustness evaluation across different depth models. The best performances of each metric are highlighted with **bold**.

| Method | Abs Rel | Sq Rel | RMSE | RMSE log | mIoU |
|---|---|---|---|---|---|
| Metric3D V2 (Hu et al., 2024) | 0.170 | 4.016 | 6.453 | 0.291 | — |
| UniDepth V2 (Piccinelli et al., 2025) | 0.158 | 2.232 | 5.491 | 0.259 | — |
| PG-Occ (Metric3D V2) | 0.139 | 1.159 | 5.466 | 0.269 | **15.15** |
| PG-Occ (UniDepth V2) | **0.131** | **1.129** | **5.049** | **0.248** | 15.08 |

## B.4 EFFECTIVENESS IN MULTIMODAL SETTINGS WITH LiDAR AND CAMERAS

While our approach targets image-based occupancy prediction, which is an important direction in the field, it is also effective in multimodal settings combining LiDAR and cameras. To validate this capability, we performed additional experiments on the nuScenes dataset, substituting pseudo-depth inputs with ground-truth sparse LiDAR point clouds. The results demonstrate that PG-Occ effectively leverages multimodal inputs, maintaining robust semantic occupancy prediction even for challenging scenes.

As summarized in 14, even a direct, naive replacement—without any tailored optimization—significantly boosts 3D occupancy metrics: mIoU rises from 15.15 (Depth) to 18.98 (Sparse LiDAR), RayIoU from 13.92 to 15.22, and mAP(v) from 21.20 to 29.53. This validates the pipeline's capability to generalize beyond pseudo-depth and effectively absorb multimodal spatial cues. Notably, our approach yields robust improvements with

Table 14: 3D occupancy performance with pseudo-depth vs. LiDAR inputs.

| Method | mIoU | RayIoU | mAP (v) |
|---|---|---|---|
| PG-Occ (Depth) | 15.15 | 13.92 | 21.20 |
| PG-Occ (LiDAR) | 18.98 | 15.22 | 29.53 |

basic point cloud substitution, indicating that further gains remain achievable via more sophisticated fusion techniques.

## C   ADDITIONAL VISUALIZATION RESULTS

### C.1   PG-OCC CAPABILITIES

We present more qualitative results in Fig. 9 demonstrating that, using single-frame multi-view inputs and feed-forward inference, PG-Occ accurately estimates scene depth and generates open-vocabulary feature renderings capturing semantics beyond fixed categories. It supports zero-shot semantic 3D occupancy prediction and enables flexible open-vocabulary text queries for object retrieval and localization.

### C.2   BEV VISUALIZATION

In this subsection, we present BEV (bird's-eye view) occupancy visualizations produced by PG-Occ. This perspective provides a comprehensive overview of the scene layout, allowing clear observation of spatial relationships among various objects. As illustrated in Fig. 10, our method accurately reconstructs both large and small scene elements, including vehicles, pedestrians, and barriers, while maintaining sharp and consistent occupancy boundaries. We select a variety of diverse scenes to demonstrate the robustness and generalization capability of our approach, highlighting its ability to handle complex environments effectively.

### C.3   EGO-CENTRIC PERSPECTIVE OCCUPANCY VISUALIZATION WITH PREVIOUS SOTA METHOD

In this subsection, we visualize occupancy from the vehicle's perspective and compare our results with the previous state-of-the-art method, GaussTR (Jiang et al., 2024). This comparison aims to highlight the strengths and improvements of our approach in estimating occupancy within the scene. As illustrated in Fig. 11, our method demonstrates superior detection results for small objects compared to GaussTR (Jiang et al., 2024), particularly for car, bicycle, bus, truck, and barrier. Interestingly, our approach is capable of detecting elements that are not well annotated in the Ground Truth, such as the pedestrians and bicycles shown in the second visualization of the figure.

### C.4   THIRD PERSPECTIVE OCCUPANCY VISUALIZATION

In this subsection, we present the visualization of our method from two different third-person perspectives. As illustrated in Fig. 12, we compare the zero-shot semantic occupancy estimations generated by our approach with the Ground Truth. The visualizations illustrate the effectiveness of our method in accurately capturing the spatial occupancy of various objects within the scene. The results underscore our model's ability to perform zero-shot semantic occupancy estimation, enabling it to infer the occupancy of objects it has not encountered during training. However, it is important to note that due to occlusion issues present in the scene, our self-supervised method may face challenges in making accurate predictions in areas lacking visual observations. Nevertheless, it can still yield reasonable inferences to a certain extent.

## D   ADDITIONAL IMPLEMENTATION DETAILS

### D.1   VOXELIZATION

As described in section 3.4, after obtaining the progressive 3D Gaussian representation that models the scene based on the current camera input. We convert the 3D feature Gaussian blobs output to a semantic occupancy field. To begin with, we take $n$ arbitrary text prompts $c_{text}$ and encode them using the CLIP text encoder to obtain their corresponding feature embeddings $f_{text}$. And then compute the similarity between these text embeddings and the text-aligned features $f_i$ of Gaussian $i$, subsequently. The text probability for each 3D feature Gaussian blob $G$ under $c_{\text{text}}$ can be calculated

Table 15: Text prompts used for zero-shot semantic occupancy estimation on the Occ3D-nuScenes dataset (Tian et al., 2024). '-' indicates that no prompts were made for this class.

| nuScenes Class | Prompts |
|---|---|
| others | - |
| barrier | barrier |
| bicycle | bicycle |
| bus | bus |
| car | car |
| construction vehicle | construction vehicle |
| motorcycle | motorcycle |
| pedestrian | person |
| traffic cone | cone |
| trailer | trailer |
| truck | truck |
| driv. surface | road |
| other flat | - |
| sidewalk | sidewalk |
| terrain | terrain, grass |
| manmade | building, wall, fence, pole, sign |
| vegetation | vegetation |
| empty | sky |

as follow equation:

$$p_i = \sigma(f_i \cdot f_{text}^T) \tag{14}$$

where $p_i$ represents the text probability of the $i$-th Gaussian blob under $c_{\text{text}}$, and $\sigma$ denotes the softmax operation.

After that, we define a voxel grid within the region of interest (ROI) occupancy range and then calculate the influence of each Gaussian on each voxel, accumulating the results. This process is affected by the anisotropy parameter $s$, $r$ of the Gaussians, their opacity $o$, and assigned text probability $p$. The formulation for this voxelization can be written as:

$$\mathcal{V}_o = \sum_{i=1}^{N} G_i(x; \mu_i, s_i, r_i, o_i) = \sum_{i=1}^{N} \exp(-\frac{1}{2}(x - \mu_i)^T \Sigma_i^{-1}(x - \mu_i))o_i, \tag{15}$$

$$\mathcal{V}_p = \sum_{i=1}^{N} G_i(x; \mu_i, s_i, r_i, o_i) = \sum_{i=1}^{N} \exp(-\frac{1}{2}(x - \mu_i)^T \Sigma_i^{-1}(x - \mu_i))p_i, \tag{16}$$

where $\mathcal{V}_o$, $\mathcal{V}_p$ denote the final occupancy probability and semantic 3D occupancy field, $x$ denotes the voxel grid position of occupancy, $\Sigma$ is the Gaussian covariance matrix of each Gaussian, revived from its scale $s_i$ and rotation quaternion $r_i$.

In the evaluation of the nuScenes retrieval dataset experiment in Section 4.2, since the ground truth consists of text annotations for sparse LiDAR points $\mathcal{P}$, we treat each LiDAR point $p$ as the center $x$ of a voxel. This allows us to obtain the corresponding final occupancy probability and text feature, as shown in the following formula.

$$\mathcal{P}_o = \sum_{i=1}^{N} G_i(p; \mu_i, s_i, r_i, o_i) = \sum_{i=1}^{N} \exp(-\frac{1}{2}(p - \mu_i)^T \Sigma_i^{-1}(p - \mu_i))o_i, \tag{17}$$

$$\mathcal{P}_f = \sum_{i=1}^{N} G_i(p; \mu_i, s_i, r_i, o_i) = \sum_{i=1}^{N} \exp(-\frac{1}{2}(p - \mu_i)^T \Sigma_i^{-1}(p - \mu_i))f_i, \tag{18}$$

where $\mathcal{P}_o$, $\mathcal{P}_f$ denote the final occupancy probability and the corresponding text feature of the LiDAR point cloud $\mathcal{P}$.

## D.2 TEXT PROMPT

Due to the imprecise semantics in the Occ3D-nuScenes (Tian et al., 2024) dataset, we made some minor adjustments to the prompts used in PG-Occ, as shown in Table 15. Specifically, we do not detect the categories 'others' or 'other flat,' as they can lead to ambiguities. Note that further fine-tuning of these ambiguous prompts could enhance performance.

For the retrieval task in Section 4.2, we directly use the prompt provided by the dataset.

## D.3 ADDITIONAL MODEL AND TRAINING DETAILS

### D.3.1 SUPERVISION STRATEGY

Metric3D V2 (Hu et al., 2024) and MaskCLIP (Zhou et al., 2022) are utilized for depth and feature supervision. The loss weight parameters are set as follows: $\lambda_{SILog} = 0.15$, $\lambda_{temp} = 10$, and $\lambda_{mse} = 10$. The learning rate is initialized at 2e-4 with a weight decay of 0.01, using the AdamW optimizer.

### D.3.2 MODEL ARCHITECTURE AND TRAINING SETUP

We adopt ResNet-50 (He et al., 2016) as the image feature backbone, utilizing the previous seven frames to capture spatio-temporal information. PG-Occ is initialized with 4,000 Gaussian queries in the base layer and progressively adds 1,000 queries per layer, resulting in one base and two progressive layers with an embedding dimension of 256. All training experiments are conducted on 8 A800 GPUs for 8 epochs, while inference is performed on a single A800 GPU. To improve computational efficiency, we use a resolution of $180 \times 320$ for depth and feature rasterization, as well as for Gaussian point initialization.

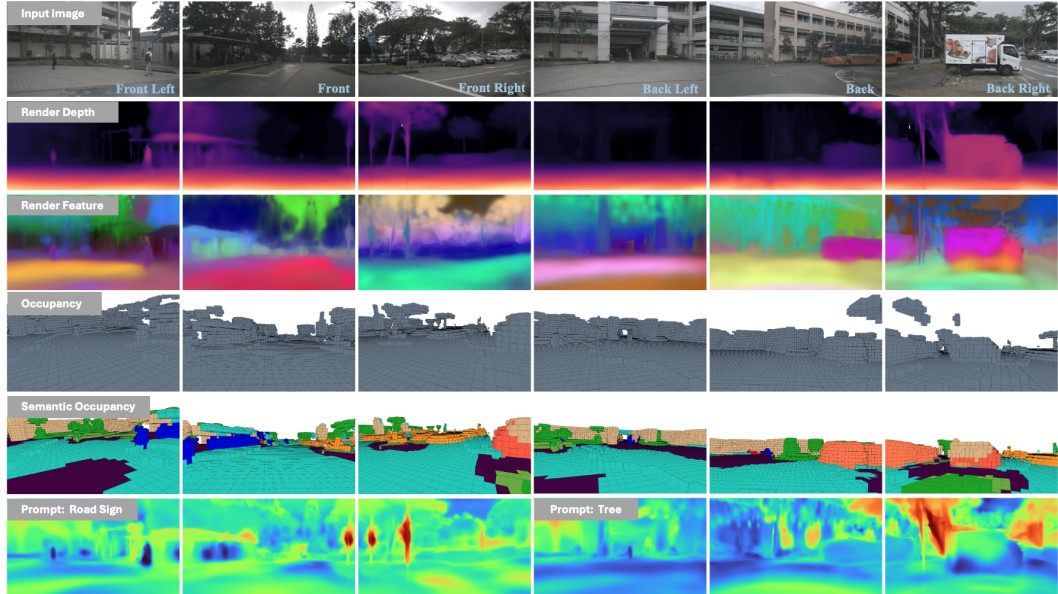

Figure 9: PG-Occ capabilities. Given only single-frame multi-view inputs and using only feed-forward passes, PG-Occ can: (1) estimate depth (row 2); (2) render open-vocabulary model features (row 3); (3) predict 3D occupancy in a zero-shot manner (rows 4); (4) predict semantic 3D occupancy in a zero-shot manner (rows 5); (5) support open-vocabulary text queries (rows 6).

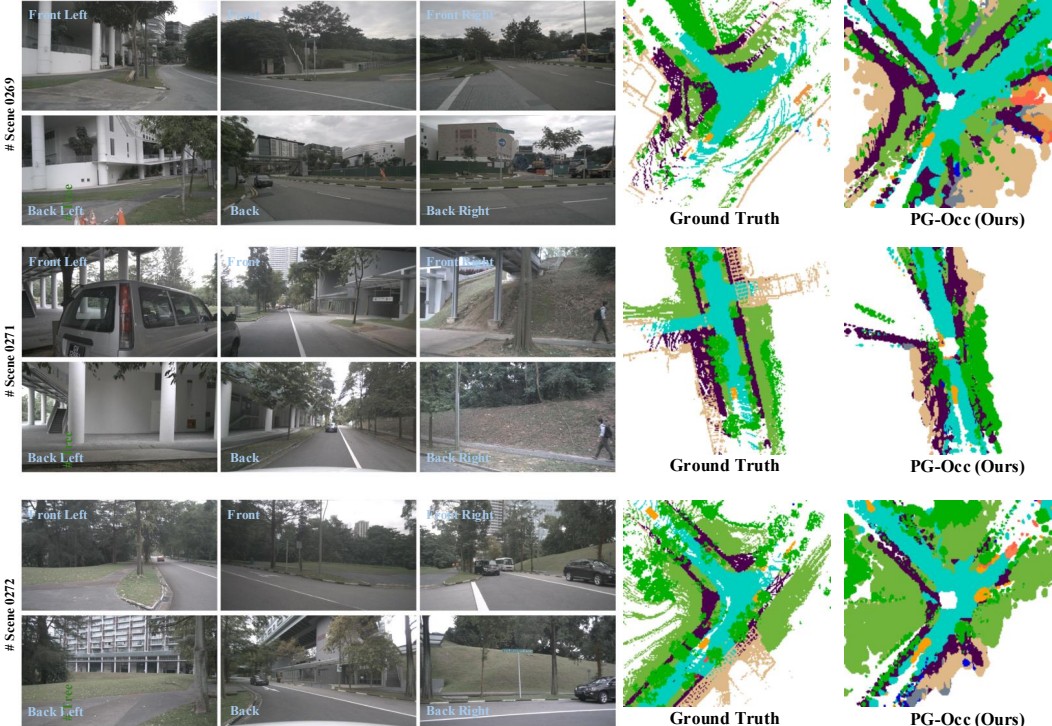

Figure 10: BEV visualization of open-vocabulary scene perception by PG-Occ. The figure illustrates predicted occupancy and semantic structures from a bird's-eye perspective, emphasizing the model's ability to capture spatial relationships and overall scene layout.

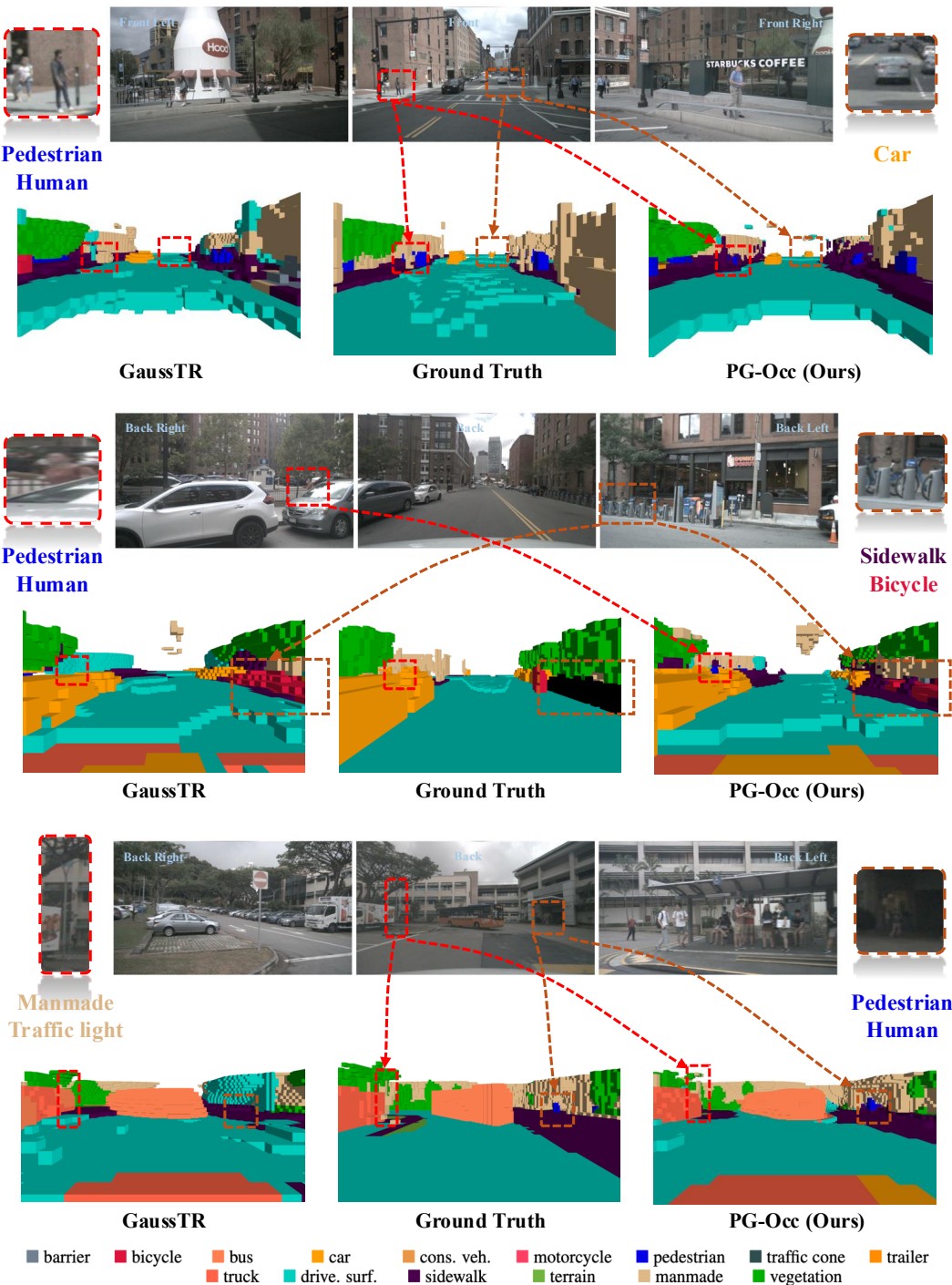

Figure 11: Qualitative comparisons of zero-shot semantic occupancy estimation from an ego-centric multi-camera perspective. Each row shows input images from multiple viewpoints (top), corresponding occupancy predictions by GaussTR (left bottom), the ground truth occupancy (middle bottom), and our PG-Occ method (right bottom). Dashed boxes and lines highlight specific objects—such as pedestrians, cars, bicycles, and traffic lights—that have been successfully detected and reconstructed. Our approach demonstrates superior detection and reconstruction of small or distant objects, better preserves spatial relationships, and provides more accurate object shapes compared with GaussTR. Colors indicate semantic categories as defined in the legend. For best inspection of fine details, we recommend viewing the color version and zooming in.

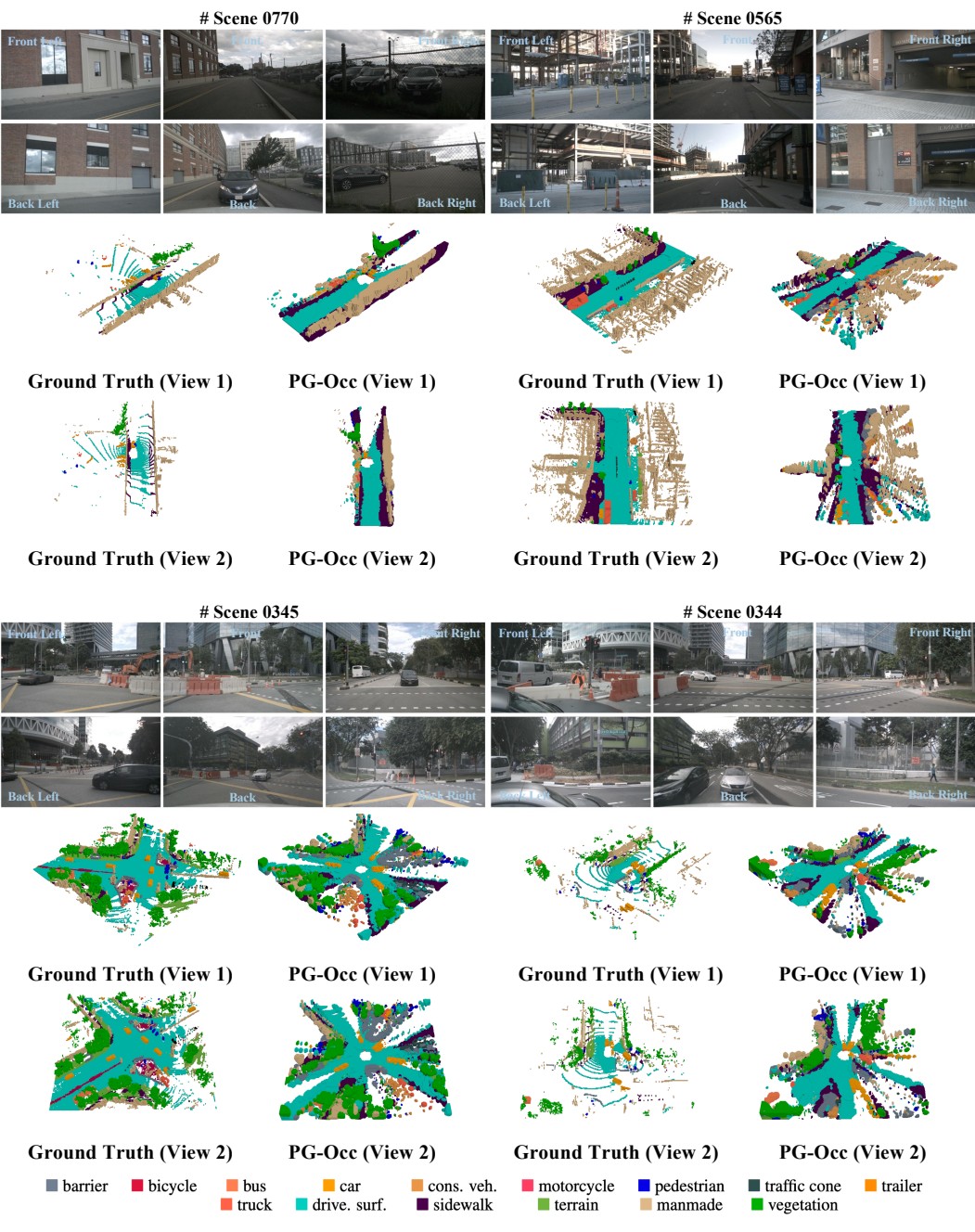

Figure 12: Qualitative zero-shot semantic occupancy results on the third perspective for two views. For each view (View 1 and View 2), we show the predictions of our method (PG-Occ) alongside the Ground Truth. The results demonstrate that PG-Occ accurately captures semantic occupancy patterns across different perspectives.

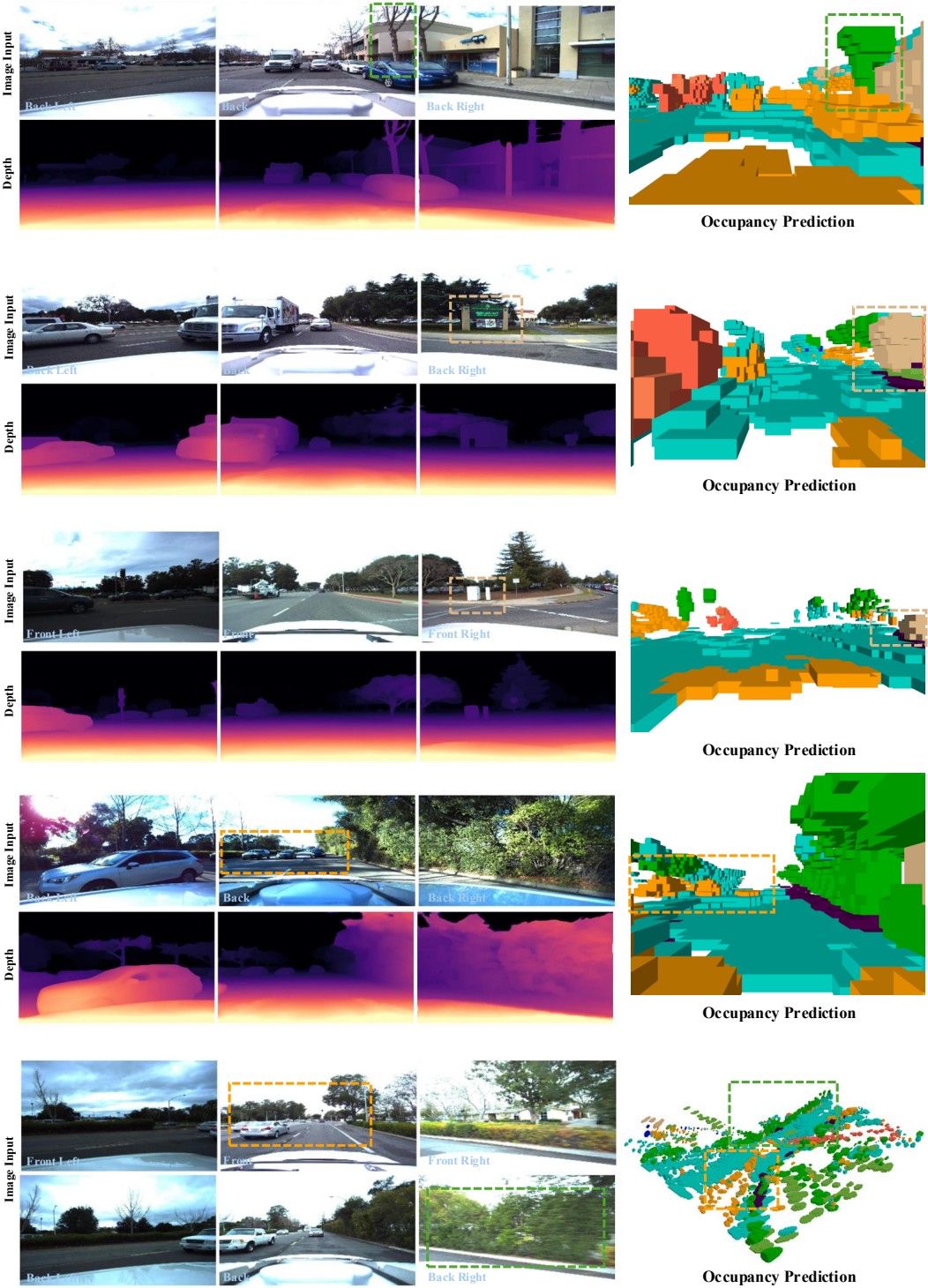

Figure 13: Zero-shot generalization on the Lyft Level-5 dataset (Christy et al., 2019). Our model is not retrained or fine-tuned on the Lyft Level-5 dataset but used directly after training on nuScenes. This scenario involves a substantial domain shift, including differences in image resolution, camera intrinsics, viewpoints, and overall scene distribution. Despite these challenges, our method maintains strong zero-shot generalization, accurately predicting occupancy and successfully recovering small or rarely seen objects in completely unseen scenes.

