# OpenReview forum: "Progressive Gaussian Transformer with Anisotropy-aware Sampling for Open Vocabulary Occupancy Prediction"
_ICLR.cc/2026/Conference — ICLR 2026 Poster_

### Official Review · Reviewer_qE6H · 2025-11-01

**Soundness:** 3
**Presentation:** 3
**Contribution:** 3
**Rating:** 6
**Confidence:** 3

**Summary:**

This paper presents PG-Occ, a Progressive Gaussian Transformer for open-vocabulary 3D occupancy prediction. The method progressively densifies 3D Gaussian representations in a feed-forward manner to capture fine scene details while maintaining efficiency. An additional anisotropy-aware sampling strategy adaptively adjusts receptive fields across scales and time for better spatio-temporal feature aggregation. Integrating language-aligned features enables text-driven 3D reasoning. Experiments show that PG-Occ achieves improvement over previous state-of-the-art methods, delivering more detailed and scalable scene understanding.

**Strengths:**

1. The paper introduces a novel Progressive Gaussian Transformer that progressively refines 3D Gaussian representations through a feed-forward densification process, balancing detail capture and computational efficiency.
2. The proposed anisotropy-aware sampling adaptively adjusts receptive fields across directions and scales, enabling more accurate feature aggregation and fine-grained geometric modeling.

**Weaknesses:**

1. Although progressive densification effectively enhances scene details, the increasing number of Gaussians at higher stages inevitably raises inference time and memory consumption. While the paper acknowledges this issue and plans to optimize it in future work, no quantitative memory analysis or profiling results are provided, leaving the practical cost-performance trade-off unclear.
2. Experiments are conducted on two benchmark datasets, which is relatively limited. The evidence for cross-domain robustness and generalization remains insufficient.
3. One of the main contributions, the anisotropy-aware sampling, provides quantitative gains in the ablation study, but the improvement is relatively small.
4. The comparisons mainly cover works published up to 2024. Recent approaches from the past year in similar or related directions are missing from the comparison.

**Questions:**

1. Could the authors provide a detailed analysis or profiling of memory consumption across different densification stages?
2. Have the authors tested the method on additional datasets or unseen domains to further validate its generalization capability?
3. While AFS shows limited quantitative improvement in the ablation, could the authors provide qualitative examples or visualizations that better illustrate its specific benefits or effects?
4. Could the authors include comparisons with more recent methods (2024-2025)?

---

> ### Author Response · Authors · 2025-11-25
>
> We sincerely thank Reviewer #qE6H for your thoughtful and constructive comments and suggestions. We also appreciate your recognition of our results and cohesive design. Based on your valuable feedback, we have further improved the manuscript and provided detailed responses below to address your concerns. (All citations refer to the revised manuscript.)
>
> **Memory Profiling Across Densification Stages:**
>
> Thank you for pointing out this aspect. We conducted a detailed memory profiling analysis, summarized in the Table below.
>
> | Densification Stage | Peak Memory Consumption (GB) |
> | --- | --- |
> | Base Stage | ~4.3 |
> | • First Progressive Stage | +0.8 |
> | • Second Progressive Stage | +1.1 |
>
> As shown, the peak memory grows gradually from 4.3 GB at the base stage to 6.2 GB at the final progressive stage, with the first densification stage introducing an additional 0.8 GB and the second adding 1.1 GB. This increase is primarily due to the growing number of Gaussian primitives (from 4,000 to 5,000 to 6,000), which increases runtime parameter storage. Despite this increase, the growth remains moderate, indicating that our progressive densification strategy scales efficiently. Overall, the results demonstrate a **controlled, gradual increase** in memory usage and confirm that the method can be executed on consumer-level GPUs, making it practical for real-world applications.
>
> **Zero-shot Generalization to Additional Datasets and Unseen Domains:**
>
> Thank you for the insightful question. To further validate the generalization capability of our method, we include additional qualitative results on the **Lyft Level-5 dataset** (see Supplementary Figure 13). Importantly, **we do not retrain or fine-tune** our model on the Lyft Level-5 dataset. Instead, we directly perform inference using the model trained solely on nuScenes.
>
> This setting represents a challenging domain shift: image resolution, camera positions, camera intrinsics, and the overall environmental domain all differ significantly from those in nuScenes. Despite these differences, our method consistently produces reliable open-vocabulary semantic occupancy predictions, and notably, it is still able to detect and recover small or rare objects in these unseen scenes.
>
> These results demonstrate that our approach generalizes effectively to new datasets and unseen domains even without any additional training.
>
> **Qualitative Benefits of Anisotropy-aware Sampling (AFS):**
>
> To further clarify the advantages of the AFS module, we provide additional qualitative results that illustrate its specific benefits in challenging scenarios. As shown in Figure 7 of our revised manuscript, AFS enables the model to capture finer, semantically-aware details, especially in regions containing complex or anisotropic geometric structures. In contrast, the model without AFS often struggles to distinguish such geometric and semantic variations. Notably, the right-side examples show that ignoring anisotropy leads to oversized Gaussian blobs near the camera, which destabilize the occupancy optimization process. By incorporating more spatial information brought by AFS, these large Gaussian blobs are effectively suppressed, resulting in cleaner and more stable occupancy predictions. These qualitative observations demonstrate the practical value of AFS, even though its quantitative gains in the ablation study appear modest.
>
> **Comparison with Recent Methods (2024-2025):**
>
> In our paper, we already included comprehensive comparisons with recent open-vocabulary 3D occupancy prediction methods. Specifically, we report results for POP-3D (NeurIPS 2023), VEON (ECCV 2024), LangOcc (3DV 2025), and GaussTR (CVPR 2025), showing that PG-Occ consistently outperforms these latest approaches.
>
> In addition to the above, we further include comparisons with GaussianFlowOcc (ICCV 2025) on both depth estimation and occupancy prediction (see revised Table 1 and Table 2). Notably, our method not only supports open-vocabulary text-query–based 3D occupancy prediction but also achieves substantial improvements on the depth estimation task, reducing Abs Rel from **0.278 to 0.139**, which highlights the effectiveness of our progressive geometry refinement pipeline.
>
> We appreciate your insightful comments and careful review of our work. If you have any further questions or suggestions, please let us know. We are glad to address them.

---

> > ### Comment · Reviewer_qE6H · 2025-11-28
> >
> > The authors’ responses effectively address my concerns. I am satisfied with the clarifications provided and am willing to raise my score accordingly.

---

> > > ### Author Response · Authors · 2025-11-28
> > >
> > > We are glad that our efforts have effectively addressed your concerns and been recognized with a higher rating, and we are sincerely grateful for your comment and support for this work.

---

### Official Review · Reviewer_TDbw · 2025-11-01

**Soundness:** 3
**Presentation:** 4
**Contribution:** 3
**Rating:** 6
**Confidence:** 3

**Summary:**

The paper introduces PG-Occ, a progressive Gaussian transformer for open-vocabulary 3D occupancy. It starts from a coarse set of 3D Gaussians and, in a feed-forward way, adds new Gaussians only where depth reveals under-modeled regions, so the scene becomes denser exactly where it needs more detail. It further makes each Gaussian anisotropy-aware and keeps the model stable with asymmetric self-attention so newly added Gaussians do not disturb existing ones.

**Strengths:**

- i) The paper targets a clear gap in current Gaussian-based open-vocabulary occupancy. The proposed progressive, depth-guided densification is a straightforward and well-motivated way to increase capacity only where the scene is under-modeled.

- ii) The anisotropy-aware feature sampling makes good use of the Gaussian’s scale and orientation, which is sensible for driving scenes where many structures are not isotropic.

- iii) The paper is well organized and clearly written.

**Weaknesses:**

- i) The whole progressive pipeline relies on the quality of the pseudo depth.  A short discussion on robustness to worse depth (or to LiDAR-sparse depth) would make the claim stronger.

- ii) Because the Gaussian set can only grow, not shrink, the last layers still have to process the largest token set. This is fine at Occ3D-nuScenes resolution, but may need pruning for HD-maps or city-scale scenes.

**Questions:**

- i) The current densification is driven by a depth discrepancy threshold. Have you tried combining this with a text-feature uncertainty signal, so that regions that are geometrically covered but semantically ambiguous can also trigger new Gaussians?

---

> ### Author Response · Authors · 2025-11-25
>
> We sincerely thank Reviewer #TDbw for your thoughtful and constructive comments and suggestions. We also appreciate your recognition of our writing, motivation, and cohesive design. Based on your valuable feedback, we have further improved the manuscript and provided detailed responses below to address your concerns. (All citations refer to the revised manuscript.)
>
> **Robustness to pseudo‑depth quality**
>
> We thank the reviewer #TDbw for pointing out the role of pseudo-depth quality. While our progressive pipeline relies on pseudo-depth inputs, it can actually produce better depth predictions than the input pseudo-depth (see Table 2, where Abs Rel improves from 0.170 to 0.139). To further demonstrate robustness, we include additional experiments in the supplementary material using LiDAR-sparse depth and pseudo-depth with Gaussian noise (standard deviations 0.1 and 0.5).
>
> For LiDAR-sparse depth, the projected sparse LiDAR points on the camera plane yield only ~3.5K valid pixels, fewer than the 4K base query points used by our baseline. As a result, the progressive component cannot fully perform iterative refinement. The observed performance degradation (mIoU from 15.15 to 14.41) is therefore expected and reflects limited input coverage rather than instability in our progressive mechanism.
>
> For Gaussian-perturbed pseudo-depth, results show that applying noise during training and inference does not significantly affect mIoU, RayIoU, or mAP, highlighting the robustness of our method to pseudo-depth quality.  In addition, following the suggestion of Reviewer #TDbw, we further evaluate the robustness of our pipeline under different pretrained depth models (see Table 13).
>
> We attribute this robustness to two main factors:
>
> (i) Occupancy prediction in nuSecene datasets operates on a 0.4m grid and does not require highly precise depth.
>
> (ii) Our feed-forward pipeline online refines Gaussian spatial parameters via spatio-temporal fusion, enabling Gaussian blobs to update to the correct occupied positions. Notably, our model often improves upon the initial Metric3D depth itself, further validating the effectiveness of online optimization.
>
> Table: Robustness to pseudo‑depth quality
>
> | Depth Source | mIoU | RayIoU | mAP (v) |
> | --- | --- | --- | --- |
> | LiDAR-Sparse Depth | 14.41 | 11.92 | 10.23 |
> | Metric3D v2 with depth noise (SD 0.1) | 15.12 | 13.95 | 21.35 |
> | Metric3D v2 with depth noise (SD 0.5) | 14.99 | 13.89 | 20.95 |
> | Metric3D v2 | 15.15 | 13.92 | 21.20 |
>
> **Combining Depth Discrepancy with Text-Feature Uncertainty for Densification:**
>
> We did not explore this in the main paper, as we mainly focus on geometry-guided feed-forward densification, but it is feasible and represents a promising direction for future research. In the supplementary material, we include a preliminary experiment incorporating a text-feature uncertainty signal. Specifically, we compute the cosine similarity between the predicted feature map and a pre-trained feature map. Pixels with similarity below 0.25 are selected as additional candidates for Gaussian densification. This introduces an extra computational overhead of approximately 25 ms per inference.
>
> As shown in the Table below, combining depth discrepancy with text-feature uncertainty slightly improves occupancy prediction compared to using depth alone. Possible reasons for the relatively modest improvement are as follows. First, most semantically ambiguous regions in our benchmark already coincide with areas of significant depth discrepancy, so the additional uncertainty-based regions contribute only marginally new supervision. Second, the text-feature similarity signal tends to be spatially sparse and noisy, limiting its effect. Finally, our feed-forward densification pipeline is primarily geometry-driven; a more principled and tightly coupled integration of geometric and semantic cues would likely be more effective and remains an interesting direction for future investigation.
>
> Table: Effect of densification strategy on 3D occupancy metrics
>
> | Densification Strategy | mIoU | RayIoU | mAP (v) |
> | --- | --- | --- | --- |
> | Depth Only | 15.15 | 13.92 | 21.20 |
> | Depth + Text-Feature Uncertainty | 15.18 | 13.83 | 21.34 |
>
> We appreciate your insightful comments and careful review of our work. If you have any further questions or suggestions, please let us know. We are glad to address them.

---

### Official Review · Reviewer_pxcU · 2025-11-01

**Soundness:** 3
**Presentation:** 4
**Contribution:** 3
**Rating:** 6
**Confidence:** 5

**Summary:**

The paper proposes PG-Occ, a progressive Gaussian Transformer framework for open-vocabulary 3D occupancy prediction. The method couples (i) Progressive densification of Gaussians through an iterative feed-forward densification strategy, (ii) Anisotropy-aware feature sampling that selects sample points and projects them onto feature planes with varying receptive fields. Extensive experimental results demonstrate that PG-Occ achieves SOTA performance on Occ3D-nuScenes.

**Strengths:**

The strengths of this paper lie in its clear motivation and cohesive design: it addresses the trade-off in Gaussian representations via progressive densification, stabilizes the training dynamics through asymmetric attention, and improves feature alignment via anisotropy that matches each Gaussian. Empirically, main results, ablations, and efficiency comparisons reinforce one another, showing that under fixed or limited compute, the approach delivers a better balance of performance and speed than previous baselines.

**Weaknesses:**

However, I also have some concerns:
(1) The robustness of corner cases is underexplored: In autonomous driving, identifying corner cases is critical. Common classes like cars, trucks, and pedestrians are already well recognized by supervised learning, whereas less common categories, such as plastic bags or trash bins, are much harder to detect. Can PG-Occ recognize such corner cases (not limited to these two examples)?
(2) From the paper, the motivation seems somewhat trivial, more like a data augmentation extension of GaussTR, so it should explicitly articulate the deeper thought behind this motivation.
(3) The method shows strong performance in a camera-based pipeline. Does the method work in a multimodal setting?

**Questions:**

1. Could you show more corner-case examples, such as uncommon categories?
2. Could you provide a deeper intuition or theoretical analysis of the motivation to justify the necessity of the method?
2. Is it also effective in a multimodal setting with LiDAR and cameras? If experiments are not feasible due to dataset constraints, an analysis of the underlying rationale would suffice.

**Details Of Ethics Concerns:**

No ethics concerns.

---

> ### Author Response · Authors · 2025-11-25
>
> We sincerely thank Reviewer #pxcU for your thoughtful and constructive comments and suggestions. We also appreciate your recognition of our motivation and cohesive design. Based on your valuable feedback, we have further improved the manuscript and provided detailed responses below to address your concerns. (All citations refer to the revised manuscript.)
>
> **More Visualization of Corner Cases:**
>
> In the Open Vocabulary Retrieval sub-figure of Fig. 1, we illustrated an example of an uncommon category, such as a garbage bin. In the revised manuscript, we have added several additional examples of uncommon categories beyond the standard classes in Fig. 7 to further demonstrate PG-Occ’s open-vocabulary capabilities. These new examples underscore the method’s ability to generalize to rarely seen objects, reinforcing its effectiveness in open-vocabulary semantic occupancy prediction.
>
> **Deeper Intuition of Our Motivation:**
>
> Our intuition stems from the challenge of efficiently capturing detailed 3D occupancy in complex driving scenes. While feed-forward sparse Gaussian representations are computationally efficient, they often fail to adapt their density to local scene complexity: globally sparse initializations tend to underrepresent tiny or intricate structures (e.g., traffic cones, trees), whereas uniformly dense initializations incur prohibitive computational costs. Vanilla 3D Gaussian Splatting demonstrates that iterative, region-aware updates can better capture local details, but it cannot operate in real time. To the best of our knowledge, no existing method achieves similar densification purely in a feed-forward manner, which motivates our approach. Accordingly, we investigate whether Gaussian densification can be realized in a fully feed-forward paradigm. Addressing this challenge is crucial for capturing fine-grained spatial details while maintaining computational efficiency, particularly in the large-scale, real-time scenarios typical of autonomous driving.
>
> **Effectiveness in Multimodal Settings with LiDAR and Cameras:**
>
> Yes, while our approach targets image-based occupancy prediction, which is an important direction in the field, it is also effective in multimodal settings combining LiDAR and cameras. This extension can be realized via:
>
> (i) LiDAR–camera feature fusion;
>
> (ii) directly replacing pseudo-depth inputs with LiDAR point clouds; or
>
> (iii) using LiDAR point clouds as semi-supervised training signals.
>
> To validate this capability, we performed additional experiments on the nuScenes dataset, substituting pseudo-depth inputs with ground-truth sparse LiDAR point clouds. The results demonstrate that PG-Occ effectively leverages multimodal inputs, maintaining robust semantic occupancy prediction even for challenging scenes.
>
> As summarized in Table 14, even a direct, naive replacement—without any tailored optimization—significantly boosts 3D occupancy metrics: mIoU rises from 15.15 (depth) to 18.98, RayIoU from 13.92 to 15.22, and mAP(v) from 21.20 to 29.53. This validates the pipeline's capability to generalize beyond pseudo depth and effectively absorb multimodal spatial cues. Notably, our approach yields robust improvements with basic point cloud substitution, indicating that further advances remain achievable via more sophisticated fusion or upsampling techniques.
>
> Table 14: 3D occupancy performance with pseudo-depth vs. LiDAR inputs.
>
> | **Method** | **mIoU** | **RayIoU** | **mAP (v)** |
> | --- | --- | --- | --- |
> | PG-Occ (Depth) | 15.15 | 13.92 | 21.20 |
> | PG-Occ (LiDAR) | 18.98 | 15.22 | 29.53 |
>
> We appreciate your insightful comments and careful review of our work. If you have any further questions or suggestions, please let us know. We are glad to address them.

---

> > ### Comment · Reviewer_pxcU · 2025-11-28
> >
> > We appreciate the authors’ responses, which effectively address my three main concerns. Therefore, I believe this is a paper worthy of acceptance, and I am maintaining my original rating.

---

### Official Review · Reviewer_BKpf · 2025-11-01

**Soundness:** 3
**Presentation:** 3
**Contribution:** 3
**Rating:** 6
**Confidence:** 4

**Summary:**

This paper introduces PG-Occ, a framework for open-vocabulary 3D occupancy prediction in autonomous driving. The central challenge it addresses is the trade-off between sparse Gaussian representations, which are efficient but miss fine-grained details, and dense representations, which incur high computational costs. It resolves this with two key contributions:
1. Progressive Online Densification (POD): A feed-forward strategy that iteratively refines the 3D Gaussian scene representation. It starts with a coarse model and progressively adds detail to regions with higher perception errors, efficiently capturing fine-grained objects without modeling the entire scene densely.
2. Anisotropy-aware Sampling (AFS): A sampling method that adaptively assigns receptive fields to Gaussians based on their specific scale and rotation (anisotropy). This allows for more effective spatio-temporal feature aggregation.

The model is trained using only 2D supervision (pseudo-depth maps and text-aligned features) without requiring 3D LiDAR data. Experiments show PG-Occ achieves state-of-the-art performance, demonstrating a significant 14.3% relative mIoU improvement over the previous best method on the Occ3D-nuScenes dataset.

**Strengths:**

1. PG-Occ introduces an efficient, online method that adaptively adds Gaussians to "under-represented regions" identified by depth errors. This allows the model to start coarse and progressively capture fine-grained details.
2. The paper originally identifies a weakness in prior methods that treat Gaussians as simple points, ignoring their shape. The AFS module is a novel solution that samples features based on the Gaussian's specific scale and rotation. This allows for adaptive receptive fields and more effective spatio-temporal feature aggregation.
3. ASA ensures training stability by allowing new, under-optimized Gaussians to learn from established ones, but not vice versa.
4. The method achieves SOTA results on the challenging Occ3D-nuScenes dataset, with a 15.15 mIoU score.

**Weaknesses:**

1. The method is trained using only 2D supervision from sparse-view cameras. This setup creates inherent ambiguity. A small, nearby object can project to a 2D feature patch similar to a large, distant object. While multi-view consistency and pseudo-depth supervision  help, they don't fully resolve this.
2. The model is initialized using pseudo-depth maps from Metric3D V2 and also supervised using them. The authors should present experimental results using other depth prediction methods to demonstrate the robustness of the proposed method.
3. The number of total queries should be provided in Table 4.

**Questions:**

1. Table 4 reports the final method's speed as 2.40 FPS. However, Table 12 reports the inference time of the final progressive layer as only 60.6 ms, and the sum of all three layers as 146.3 ms (27.4 + 58.3 + 60.6). This sum suggests a throughput of ~6.8 FPS. What component is responsible for the major bottleneck that reduces the final speed to 2.40 FPS? Is it the ResNet-50 spatio-temporal backbone, the final Gaussian-to-voxel post-processing, or another part not listed in Table 12?
2. You acknowledge a key limitation: "constraining the Gaussian scale in depth is challenging, which can cause popping artifacts". Could you elaborate on the severity and frequency of these artifacts? For example, do they primarily affect distant/occluded objects, or is it a general instability? How much do you believe this instability impacts the method's reliability for downstream tasks like motion planning, which require temporally stable geometric representations?
3. The model is initialized and supervised by pseudo-depth. Have you investigated the model's robustness under different depth models?
4. The ablation study shows that removing ASA ("w/o ASA") degrades performance. Did you also experiment with using standard, symmetric self-attention instead of just removing it?

---

> ### Author Response · Authors · 2025-11-25
>
> We sincerely thank Reviewer #BKpf for your thoughtful and constructive comments and suggestions. We also appreciate your recognition of our contributions and results. Based on your valuable feedback, we have further improved the manuscript and provided detailed responses below to address your concerns. (All citations refer to the revised manuscript.)
>
> **Clarification on Computational Bottleneck and FPS Discrepancy:**
>
> The main reason for the FPS drop to 2.40 is the Gaussian-to-Voxel (voxelization) module, which constitutes the dominant computational bottleneck in our pipeline. This conversion alone takes approximately 224 ms. It is not included in Table 12, as that table primarily reports layer-wise time consumption. The runtime of the voxelization step depends not only on the total number of Gaussians but also on their anisotropic properties (e.g., rotation and scale) and the resolution of the target occupancy grid. This overhead can be mitigated in two ways: (1) by pruning uninformative Gaussians, such as those with low opacity or not visible in the scene, or (2) by using Gaussian primitives directly for downstream perception tasks, which eliminates the need for voxel conversion.
>
> **Clarification on Popping Artifacts:**
>
> The “popping” artifacts introduced by imperfect depth-direction constraints are generally mild and occur infrequently. In practice, such artifacts primarily emerge in **unobservable or heavily occluded regions**, where geometric supervision is inherently ambiguous. For example, slight shape stretching of vehicles can occasionally be observed (e.g., the elongated car in Scene 0344 of Figure 12). In contrast, the geometry within camera-visible regions—where depth cues are reliable—remains temporally stable.
>
> Importantly, the limited and localized nature of these artifacts results in **minimal influence on downstream tasks**, including motion planning. As shown in the ego-perspective visualizations in Fig. 9 and the supplementary demo videos, the predicted occupancy in the visible region is temporally consistent. In real-world pipelines, regions that are occluded or poorly constrained can be naturally treated as high-uncertainty zones by downstream modules, preventing them from affecting safety-critical decisions.
>
> Overall, while depth direction ambiguity can theoretically introduce popping artifacts, its severity and frequency are low, it is confined to poorly observed areas, and it does not compromise the reliability of PG-Occ for downstream tasks requiring stable geometric representations.
>
> **Robustness Evaluation with Different Pretrained Depth Models:**
>
> In the main paper, we only adopt Metric3D V2 (Hu et al., 2024) as our depth estimator, as our technical contribution is independent of which pretrained depth model is utilized. To further assess the robustness of our approach under different depth estimators, we additionally train and evaluate PG-Occ using UniDepth V2 (Piccinelli et al., 2025). Table 13 reports both depth estimation errors and open-vocabulary semantic occupancy performance on the nuScenes validation set.
>
> Similar to the setting with Metric3D V2 pseudo-depth, using UniDepth V2 also enables PG-Occ to recover depth estimates that outperform the pseudo-labels. For example, while UniDepth V2 provides an Abs Rel of 0.158, PG-Occ improves it to 0.137, showing that PG-Occ can consistently refine imperfect depth supervision regardless of the depth model used.
>
> In terms of occupancy prediction, the performance remains stable across depth models. When switching from Metric3D V2 to UniDepth V2 pseudo-depth, the mIoU only changes slightly from 15.15 to 15.08, confirming that PG-Occ is largely insensitive to the specific choice of depth estimator.
>
> These results highlight two key properties of PG-Occ.
>
> - First, the method does not rely on any specific depth architecture; it only requires coarse geometric cues for initialization and supervision, making it naturally compatible with a wide range of metric depth models.
> - Second, PG-Occ can consistently refine these depth cues and maintain robust occupancy performance even when the upstream depth model changes.
>
> Table 13: Robustness evaluation across different depth models.
>
> | **Method** | **Abs Rel** | **Sq Rel** | **RMSE** | **RMSE log** | **mIoU** |
> | --- | --- | --- | --- | --- | --- |
> | Metric3D V2 | 0.170 | 4.016 | 6.453 | 0.291 |  |
> | UniDepth V2 | 0.158 | 2.232 | 5.491 | 0.259 |  |
> | PG-Occ (Metric3D V2) | 0.139 | 1.159 | 5.466 | 0.269 | 15.15 |
> | PG-Occ (UniDepth V2) | 0.131 | 1.129 | 5.049 | 0.248 | 15.08 |

---

> ### Author Response · Authors · 2025-11-25
> **Extended Official Comment by Authors**
>
> **Clarification on the ASA ablation study:**
>
> Yes, we have evaluated the use of standard, symmetric self-attention. In our original ablation study (Table 3), “w/o ASA” referred to replacing our asymmetric self-attention (ASA) module with the standard symmetric self-attention (SA). We thank the reviewer #BKpf, noting that this terminology was ambiguous. To provide a clearer comparison, we have refined the naming and additionally introduced an experiment that removes self-attention entirely. In the updated ablation, “w/o SA” denotes removing all self-attention, “w/o ASA” denotes using standard symmetric self-attention, and “Full model” corresponds to using all of our proposed modules.
>
> We evaluate all three main variants on mIoU, RayIoU, and mAP(v). The results show that while standard self-attention (SA) improves performance over the no-attention baseline, ASA consistently achieves the highest accuracy across all metrics. In particular, the asymmetric design improves mIoU from 14.85 to 15.15. This suggests that, beyond enabling cross-Gaussian communication, ASA more effectively preserves historical features while suppressing interference from newly added Gaussian components during progressive densification. Consequently, ASA serves as a more reliable feature aggregation mechanism for our task and leads to higher overall accuracy.
>
> Table 3: Effects of the proposed key model components (i.e., POD, AFS, ASA).
>
> | **Method** | **mIoU** | **RayIoU** | **mAP (v)** |
> | --- | --- | --- | --- |
> | w/o POD | 14.84 | 12.58 | 19.21 |
> | w/o AFS | 15.03 | 13.56 | 20.12 |
> | w/o SA | 11.14 | 10.44 | 15.60 |
> | w/o ASA | 14.85 | 12.76 | 19.41 |
> | Full model | 15.15 | 13.92 | 21.20 |
>
> We appreciate your insightful comments and careful review of our work. If you have any further questions or suggestions, please let us know. We are glad to address them.

---

### Meta-Review · Area_Chair_b61M · 2025-12-16

**Summary:**

This paper proposes a Progressive Gaussian Transformer framework for open-vocabulary 3D occupancy prediction, reviewers agree the proposed method is novel (BKpf, qE6H) and clearly motivated (pxcU, TDbw), the results are SOTA (BKpf, pxcU), the manuscript is well-written (TDbw).

Especially, several reviewers agree the approach delivers a better balance of performance and speed than previous baselines by progressive densification, stabilizing the training dynamics through asymmetric attention, and improving feature alignment via anisotropy that matches each Gaussian.

**Reviewer Concerns:**

Most of the concerns are addressed by the rebuttal, while below ones still hold:

The method is trained using only 2D supervision from sparse-view cameras. This setup creates inherent ambiguity. A small, nearby object can project to a 2D feature patch similar to a large, distant object. While multi-view consistency and pseudo-depth supervision help, they don't fully resolve this. (BKpf)

Because the Gaussian set can only grow, not shrink, the last layers still have to process the largest token set. This is fine at Occ3D-nuScenes resolution, but may need pruning for HD-maps or city-scale scenes. (TDbw)

Authors did not provide response for those weaknesses.

**Reviewer Scores:**

BKpf may keep 6 or increase the score.
pxcU keeps his score 6.
TDbw may keep 6 or increase the score.
qE6H is willing to raise his score.

---

### Decision · Program_Chairs · 2026-01-26

Accept (Poster)